

# Impacts of the July 2012 Siberian Fire Plume on Air Quality in the Pacific Northwest

Andrew Teakles[1], Rita So[2], Bruce Ainslie[2], Robert Nissen[2], Corinne Schiller[2], Roxanne Vingarzan[2], Ian McKendry[3], Anne Marie Macdonald[4], Daniel A. Jaffe[5,6], Allan K. Bertram[7], Kevin B.Strawbridge[4], W. Richard Leaitch[4], Sarah Hanna[7], Desiree Toom[4], Jonathan Baik[2], Lin Huang[4]

[1]Meteorological Service of Canada, Environment and Climate Change Canada, Dartmouth, NS, Canada
[2]Meteorological Service of Canada, Environment and Climate Change Canada, Vancouver, BC, Canada
[3]Department of Geography, the University of British Columbia, Vancouver, BC, Canada
[4]Science and Technology Branch, Environment and Climate Change Canada, Toronto, ON, Canada
[5]School of Science, Technology, Engineering, and Mathematics, University of Washington Bothell, Bothell, WA, USA
[6]Department of Atmospheric Sciences, University of Washington, Seattle, WA, USA
[7]Department of Chemistry, the University of British Columbia, BC, Canada

*Correspondence to*: Andrew D. Teakles (Andrew.Teakles@canada.ca)

**Abstract.** Biomass burning emissions emit a significant amount of trace gases and aerosols and can affect atmospheric chemistry and radiative forcing for hundreds or thousands of kilometers downwind. They can also contribute to exceedances of air quality standards and have negative impacts on human health. We present a case study of an intense wildfire plume from Siberia that affected the air quality across the Pacific Northwest on July 6-10, 2012. Using satellite measurements (MODIS True Colour RGB imagery and MODIS AOD), trajectories, and dispersion modelling, we track the wildfire smoke plume from its origin in Siberia to the Pacific Northwest where subsidence ahead of a subtropical Pacific High made the plume settle over the region. The normalized enhancement ratio of $O_3$ and $PM_1$ relative to CO of 0.26 and 0.09 are consistent with a plume aged 6-10 days. The aerosol mass in the plume was mainly submicron in diameter ($PM_1/PM_{2.5}$ = 0.97) and the part of the plume sampled at the peak of Whistler Mountain was 87% organic material. Stable atmospheric conditions along the coast limited the initial entrainment of the plume and caused local anthropogenic emissions to buildup. A synthesis of air quality from the regional surface monitoring networks describes changes in ambient $O_3$ and $PM_{2.5}$ during the event and contrasts them to baseline air quality estimates from the AURAMS chemical transport model without wildfire emissions. Overall, the smoke plume contributed significantly to the exceedances in $O_3$ and $PM_{2.5}$ air quality standards and objectives that occurred at several communities in the region during the event. Peak enhancements in 8-hr $O_3$ of 34-44 ppbv and 24-hr $PM_{2.5}$ of 14-32 μg/m$^3$ were attributed to the effects of the smoke plume across the Interior of British Columbia and at the Whistler Peak high elevation site (2182m ASL). Lesser enhancements of 10-12 ppbv for 8-hr $O_3$ and of 4-9 μg/m$^3$ for 24-hr $PM_{2.5}$ occurred at Whistler Peak and across coastal British Columbia and Washington State. The findings suggest that the large air quality impacts seen during this event were a combination of the efficient transport of the plume across the Pacific, favorable entrainment conditions across the BC interior and the large scale of the Siberian wildfire emissions. A warming climate increases the risk of increased wildfire activity and events of this scale re-occurring under appropriate meteorological conditions.





**Keywords:** long-range atmospheric transport, smoke, enhancement ratio, threshold exceedance, ozone, particulate matter

## 1 Introduction

Wildfires emit significant amounts of primary pollutants including fine particulate matter ($PM_{2.5}$), carbon monoxide (CO),
nitrogen oxides ($NO_x$), and non-methane hydrocarbons (NMHCs). These pollutants can react to form ozone ($O_3$) and
secondary organic aerosols (SOA), which can affect downwind air quality and climate over a wide range of scales from local
(Akagi et al., 2011, 2012, 2013; Smolyakov et al., 2014), to regional (Wigder et al., 2013), to inter-continental (Jaffe et al.
2001, 2004; Bertschi et al., 2004; Bertschi and Jaffe, 2005; Lapina et al., 2006; Pfister et al., 2006; Weiss-Penzias et al.,
2007; Bein et al., 2008; Strada et al., 2012; Jaffe and Wigder, 2012). Short-term exposures to air pollution from biomass
burning have been found to be associated with a range of health impacts, including respiratory symptoms, increased hospital
admissions and emergency room visits and premature mortality (Henderson and Johnston, 2012).

The rate of $O_3$ and SOA formation in wildfire plumes, and hence the potential for adverse air quality and health impacts, can
vary considerably due to several factors, including biomass fuel type, combustion characteristics, atmospheric conditions,
and downwind distance (Akagi et al., 2011; Jaffe and Wigder, 2012). The $O_3$ production in biomass burning is driven by
photochemical reactions of $NO_x$ and NMOCs and typically occurs in a $NO_x$ limited environment. $O_3$ enhancements relative
to CO, also known as the normalized enhancement ratio (NER; $\Delta O_3 / \Delta CO$), for temperate and boreal wildfires generally
increase as the plume ages and is typically in the range of 0.1 to 0.7 (Jaffe and Wigder, 2012). A similar trend is present for
the NER of $PM_1$ ($\Delta PM_1 / \Delta CO$) for plumes aged <2 days; yet for older plumes, SOA formation can be overshadowed by PM
loss due to wet and dry deposition, cloud processing and/or evaporation (Wigder et al., 2013). Additional studies on long-
range transport (LRT) smoke events from wildfires can help better understand the interaction of the above factors on
downwind air quality.

The Russian boreal forests represent the largest forested region on Earth with an aerial coverage of approximately 8 million
km$^2$ (Stocks, 2004). About 1% of this region is damaged from 10,000 to 35,000 forest fires annually (Isaev et al., 2002).
Jaffe et al. (2004) estimated that the active Siberian wildfire season in 2003 caused summertime $O_3$ enhancements of 9-17
ppbv, which led to episodic $O_3$ exceedances in western North America. Other studies have also demonstrated the LRT of
wildfire smoke from Siberia can affect air quality over Asia (Jeong et al., 2008; Jung et al., 2016) and contribute to artic haze
(Brock et al., 2011).

This study examines an episodic LRT event of Siberian wildfire smoke to the Pacific Northwest that had significant and
widespread air quality impacts on July 6-10, 2012. Record temperatures and dry weather across Siberia in 2012 led to
increased wildfire activity. Over 17,000 wildfires were detected in July and August alone. The Fire Inventory from NCAR
model (FINN; Wiedinmyer et al., 2011) estimated that 48 Tg of CO emissions occurred by August which was already more



than double the biomass burning emission for the entire 2010 season and second only to the 72 Tg from the 2003 season (NASA, 2012).

Cottle et al. (2014) examined the impact of this event in the Lower Fraser Valley (LFV) in British Columbia, Canada using lidar measurements taken at the University of British Columbia. Their findings suggest that the high $PM_{2.5}$ observed by the region's fixed air quality monitoring network was associated with periods of noted entrainment signatures in lidar measurements. This study expands on the Cottle et al. (2014) work in a number of significant ways: it analyzes potential air quality impacts over a much greater geographical area encompassing large part of British Columbia and Washington State; it uses detailed air quality measurements at a high elevation background site to provide insight into plume chemistry; and it makes use of photochemical modelling to establish baseline air quality conditions in the absence of any wildfire emissions to better quantify the impacts of the plume on $O_3$ and $PM_{2.5}$ levels.

The objectives of this study are 1) to characterize the trans-Pacific advection of the 2012 Siberian fire plume and its eventual entrainment into the local valleys of British Columbia and northwest Washington State, 2) to determine its contribution to degraded air quality and exceedances of regional air quality objectives and national standards and 3) to provide information on the chemical composition of the smoke plume.

## 2. Methods

### 2.1 Monitoring Data

#### 2.1.1 Ambient Air Quality Monitoring Data

Ambient $O_3$ and $PM_{2.5}$ data were obtained from several air quality monitoring networks (Fig. 1 b and c), including 35 stations from the British Columbia Ministry of Environment (BCMoE) network (BC MoE, 2014), 22 air quality monitors operated by Metro Vancouver (Metro Vancouver, 2014) within the Lower Fraser Valley (LFV), and 15 stations from the Washington State Monitoring Network (WSMN) (WA ECY, 2012). The air quality monitoring network across British Columbia is, in part, co-managed by the National Air Pollution Surveillance Network (NAPS) (Dabek-Zlotorzynska et al., 2011). Study sites also included two marine background sites (Amphitrite Point, Ucluelet, 18 m ASL on Vancouver Island (McKendry et al., 2014); and Cheeka Peak, 480 m ASL in Washington State (Weiss-Penzias et al., 2004)) and two elevated background sites (Whistler Peak, 2182 m ASL, located 120 km north of Vancouver (Takahama et al., 2011; Macdonald et al., 2011); and Mt. Rainier Jackson Visitor Centre, 1782 m ASL, located in Pierce County, Washington State). A description of the data analyzed and sampling methodologies for the various networks and sampling sites is summarized in Table 1.



### 2.1.2 Lidar Sites

Two lidars were used in this study: one located on the University of British Columbia Point Grey campus in Vancouver and a second located at the north end of the Whistler village (650 m ASL), approximately 8 km north of the Whistler Peak High Elevation (WHI) (Gallagher et al., 2012; Strawbridge, 2013). The lidars use a Continuum Inlite III laser with an approximate

output power of 1.5W at 10 Hz and operates simultaneously at wavelengths of 532 and 1064 nm. The system measures the return signals through 3 detection channels: one at 1064 nm, and two at 532 nm (one for each polarization). Backscatter information provided by the upward pointing setup has a vertical resolution of 3 m from near the ground up to 15 km and uses a 10s averaging period. Additional details on the lidar systems can be found in Strawbridge (2013).

### 2.1.3 Whistler Peak High Elevation site

Air quality measurements were collected at the Whistler Peak High Elevation site (see Table 1 for details). CO was measured with Thermo Environmental Instruments Inc., Model 48C-Trace Level analyzer and $O_3$ was measured with a Thermo Environmental Instruments Inc., UV absorption monitor (TECO 49C). Particle chemistry was provided both through integrated filter samples and continuously with an Aerosol Chemical Speciation Monitor (ACSM). Inorganic ions ($Cl^-$, $NO_3^-$, $Na^+$, $NH_4^+$, $K^+$, $Mg^{2+}$, and $Ca^{2+}$) were analyzed on 48-hour filter samples by ion chromatography (IC) and ICP-

MS. OC, POC, and EC measurements were also obtain on a Sunset lab OC-EC aerosol analyzer (http://www.sunlab.com) using the EnCan Total-900 thermal method (Huang et al., 2006) with an 8 day sampling period (July 3-11) split between day and night. The ACSM provided particle chemical composition for $SO_4^{2-}$, $NO_3^-$, $Cl^-$, $NH_4^+$, organics on a 30 min time resolution. Optical measurements of black carbon (rBC) were acquired using the Single Particle Soot Photometer (SP2). During the study, the SP2 instrument had a steady 6% loss in its laser power; however internal diagnostics suggest that the

power loss is unlikely to significantly affect the readings because heating aerosol during intake was unaffected. SP2 readings were also affected by a leak in the inlet plumbing noticed after the event. As a result, hourly estimates of EC were estimated by calibrating the rBC values from the SP2 to the 8 day integrated EC obtained from the Sunset lab OC-EC aerosol analyzer. Particle size distributions were measured by an optical particle counter (OPC) over the (0.25 to 32 um) for the entire period and also from a Scanning Mobility Particle Sizer (SMPS) (14-572 nm diameter) from July 9 onwards.

Details of the trace gas and particle measurements at Whistler Peak are discussed by Macdonald et al. (2011) and Takahama et al. (2011).

### 2.2 Trajectory and Dispersion Modelling

A detailed analysis by Cottle et al. (2014) using HYSPLIT (Hybrid Single-Particle Lagrangian Integrated Trajectory) trajectories and high resolution smoke dispersion models generated with the Bluesky framework (Sullivan et al., 2008)

concluded that wildfire sources within North America did not significantly influence air quality within the LFV during the July 6-10 period.



A similar analysis using trajectory modelling was done using the CMC Trajectory Model (D'Amours et al., 2015) based on GEM-LAM 15 km meteorological fields at 15 minute intervals. Forward trajectories were integrated for a 72 hour period at release heights of 10 m, 100 m, and 1000 m AGL. Particle dispersion runs were calculated using the HYSPLIT model Version 4 (Stein et al., 2015) with WRF (Weather Forecasting and Research) model using meteorological fields at a 35 km resolution. Particles were released at heights of 10 m, 2.5 km, 5 km, and 7.5 km AGL then tracked for period of 120 hours.

## 2.3 Analysis of $O_3$ and $PM_{2.5}$ impacts

Based on measurements collected at 72 air quality monitoring stations in the Pacific Northwest, the air quality impacts of the Siberian wildfire plume were assessed by examining the $O_3$ and $PM_{2.5}$ concentrations at various averaging periods of 1-hr, 8-hr and 24-hr, depending on the pollutant. Using the regional objective and national standards, applicable at the time of the event (Table 2), as benchmarks, severe air quality episodes were identified and compared to the average historical July daily maxima values from 2000 to 2010 at each of the monitoring stations. It should be noted that some of these standards, for example the CWS for $O_3$ (CCME, 2014), are based on multi-year statistics and were not considered in this study.

To further characterize the timing and spatial extent of these impacts, baseline concentrations of $O_3$ and $PM_{2.5}$ over the Pacific Northwest, in the absence of any wildfire emissions from either Siberia or from within North America, were simulated using the AURAMS (A Unified Regional Air-quality Modelling System) air quality transport model (Gong et al., 2006 with updates from Kelly et al., 2012) and were compared with observations. The reliability of the AURAMS baseline simulation was determined by examining the range of differences between observed and modelled values during non-event days (July 5 and from July 12-16). This range was used to estimate the uncertainty in LRT enhancement (observed – baseline) at each monitoring location. For reporting purposes, enhancements for multiple monitoring locations were aggregated, when appropriate, to reflect the average contribution of the Siberian plume across the different regions in the Pacific Northwest. A brief summary of the AURAMS model configuration is described below.

AURAMS was run over a 12 day period (July 5-16, 2012) using a nested configuration of 12- and 4-km grid spacing with the inner (4-km) domain covering southern British Columbia and northern Washington State (Fig. 1 a and b). Meteorology for the simulations was provided by Environment and Climate Change Canada's (ECCC) Global Environmental Multiscale (GEM) Limited-Area Model (LAM) weather forecast model (Côté et al., 1998) run at a 2.5-km resolution and then interpolated to the AURAMS domains. GEM-LAM was run in a series of 30-hour simulations starting from 00Z with the first six hours of each simulation discarded as "spin-up" in order to allow model fields to reach steady-state. The model was run with anthropogenic emissions based on the 2010 Canadian and 2008 U.S. emission databases. Emission totals for both databases were adjusted to 2012 levels using Metro Vancouver forecasted and backcasted LFV emission estimates (GVRD, 2007). The model used biogenic emissions that calculated using the Biogenic Emissions Inventory System (BEIS) version 3.0.9 emissions algorithms and Biogenic Emissions Landuse Database 3 (BELD3) data. AURAMS employed the ADOM-II





gas-phase chemical mechanism (Stockwell et al., 1989) and used a chemically-speciated 12-bin sectional distribution to characterize particulate matter.

Lateral boundary conditions to the AURAMS model along its outer (12 km) domain were supplied by seasonally representative climatological values for some species and for $O_3$ an additional adjustment was made to the climatological boundary conditions in response to the local tropopause height (Makar et al. 2010). Lateral boundary conditions to the inner 4 km domain were supplied from the 12 km model output. AURAMS runs did not include any wildfire emissions from either Siberia or from within North America.

### 2.4 Analysis Software

The majority of the data analyses in this study were developed in the R Language (R Core Team, 2015) and the ggplot2 graphing libraries (Wickham, 2009). The Skew-T plots of atmospheric radiosonde data were plotted in python using the SHARPpy package (Halbert et al., 2015).

## 3. Results and Discussion

### 3.1 Trans-Pacific transport

On June 29[th], 2012, smoke originating from wildfires over Eastern Russia, as noted by the NASA Earth Observatory group, drifted eastward across the Sea of Japan, the Sea of Okhotsk, and Kamchatka (Fig. 2 a). Moderate Resolution Imaging Spectroradiometer (MODIS) Aerosol Optical Depth (AOD) imagery between July 1[st] and July 6[th], shown in Fig. 2 (b, c, d), illustrates the progression of the aerosol plume (AOD > 1) across the Pacific to western North America. Long-range transport of aerosols was aided by strong zonal flow established as a developing storm tracked from the Bearing Sea to the Gulf of Alaska. Subsidence ahead of the subtropical Pacific high is thought to have contributed to the descent of the plume over the study area. In western North America, subsidence and mountain wave activity are found to be important factors in bringing mid-tropospheric dust layers in range of boundary layer entrainment process (McKendry et al., 2001; Hacker et al., 2001). Five-day forward and backward HYSPLIT dispersion modelling from Yakutsk (YKS), Russia and Vancouver (YVR), Canada, respectively, support an estimate of plume aged > 6 days upon arrival. Air parcels lofted between 5 km and 7.5 km (Fig. S1) reach the Pacific Northwest consistent with the MODIS AOD data and backward dispersion results, indicating air parcels at 5 km ASL would trace back to Asia on July 1[st].

### 3.2 July 6-10 smoke event overview

On July 7[th], the MODIS True Colour imagery (Fig. 3) shows the smoke plume oriented northeast to southwest across the southern British Columbia and Washington State. At this time, degraded air quality conditions were observed inland over the Interior of British Columbia, where both the 8-hr $O_3$ and 24-hr $PM_{2.5}$ exceedances occurred at Kamloops (KFS). From





July 8-9, poor air quality conditions continued to spread across the Interior, coinciding with the northward and inland progression of the plume, and caused air quality exceedances at several communities (see Table 3). A forward trajectory analysis (Fig. S2) from nearby North American wildfires (Long Draw, Oregon and Waldo Canyon, Colorado) confirmed that the smoke from these sources were unlikely to have contributed to the degraded ambient air quality observed. Over in the coastal region, elevated $O_3$ and $PM_{2.5}$ also occurred on July 8th, with isolated 8-hr $O_3$ exceedances at Chilliwack (CA) and Enumclaw (ENC). MODIS AOD show the plume remnant had pushed back southward on July 10th ahead of a trough of low pressure over the Interior of British Columbia (L2 on Fig. S3).

Figure 4 provides an overview of the maximal enhancements to 8-hr $O_3$ and 24-hr $PM_{2.5}$ estimated based on the differences between the ambient air quality observation and the baseline AURAMS air quality modelling for the event. Overall, the baseline modelling suggests that the greatest air quality impacts from the Siberian wildfire smoke were in the Interior of British Columbia and at the Whistler High Elevation monitoring site with peak increases of 34-44 ppbv for 8-hr $O_3$ and 14-32 μg/m$^3$ for 24-hr $PM_{2.5}$. The coastal portions of British Columbia and Washington State saw lesser enhancements of 10-12 ppbv and 4-9 ug/m$^3$, respectively. The regional disparity in the enhancement may be attributable to difference in subsidence across the region. Coastal regions had stable atmospheric conditions, leading to stagnant conditions under a strong thermal inversion between 500 and 1000 m ASL (Fig. 5a). Stable atmospheric conditions likely limited the entrainment of the plume over the coastal area and caused a buildup of local anthropogenic emissions. Over the Interior of British Columbia, dry and less stable conditions existed with daytime mixing heights over 3 km ASL (Fig. 5b). The variations in 8-hr $O_3$ and 24-hr $PM_{2.5}$ enhancements (with uncertainty estimates) across the Pacific Northwest are listed in Table 4 and are discussed in more detail in the following sections.

## 3.3 Whistler

The Whistler Peak High Elevation air chemistry monitoring site is situated to monitor free tropospheric background air pollutants and to study long-range transport events across the Pacific (Leaitch et al., 2009; Macdonald et al, 2011). On July 6th, significant increases in 1-hr $O_3$, CO, and $PM_{2.5}$ mark the onset of the smoke event at Whistler Peak (Fig. 6) and coincide with the timing of elevated aerosol layers detected by the Whistler lidar. Aerosol backscatter ratios show that the plume persisted over Whistler from July 6 14:00 PST (WHI1) to July 8 06:00 PST (WHI2) at an elevation of 2-3.6 km ASL. Backscatter ratios also indicate the entrainment of the aerosols deeper into the Whistler Valley as hourly observed $O_3$, $PM_{2.5}$, and CO reached peaks of 86 ppbv, 30 μg/m$^3$, 276 ppbv, respectively. The maximum 1-hr $O_3$ and CO values were approximately 50 and 167 ppbv above the average July background levels as reported in Macdonald et al. (2011) for 2002-2006. $O_3$ conditions prior to the event were consistent with July background levels.

The ratio of $O_3$ to CO enhancements ($\Delta O_3/\Delta CO$) provides some insight into the relative photochemical production of $O_3$ in wildfire plumes (Jaffe and Wigder, 2012) and is strongly influenced by the travel time between the fire and measurement location (Mauzerall et al., 1998). For this study, the plume age is estimated between 7-10 days old based on a Siberian origin of June 29th, 2012. The $\Delta O_3/\Delta CO$ ratio was estimated using observations from WHI1 to WHI2 and excluding cases





where CO was lower than the July background (110 ppbv). The regression slope of $O_3$ and CO for the event ranged from 0.15 to 0.24 with a mean value of 0.26 (n= 20, $r^2$= 0.80) and is comparable to other similarly aged Siberian plumes with excess ratios of 0.22-0.36 (6-10 days; Bertschi et al., 2004) and 0.15-0.84 (7-10 days, n=5; Bertschi and Jaffe, 2005).

As shown in Figure 6d, during the smoke event (WHI1 to WHI2), the $PM_{2.5}$ mass was dominated by submicron aerosol with

an average $PM_1/PM_{2.5}$ ratio of 0.97. The particle composition as determined from the ACSM (Fig. 7) during the event was on average approximately 87% organic mass, 4% $NO_3^-$, 6% $SO_4^{2-}$, 3% $NH_4^+$, 0.3% $Cl^-$. $PM_1$ mass is estimated from the ACSM, since the size distribution data from 10 nm to 10 μm indicate that most of the mass is below 0.7 μm; in Figure S4, an example is shown for July $9^{th}$, 2012 when the SMPS became operational again. Further, a comparison of the mass concentrations estimated from the SMPS with the ACSM for July 9-31 (Figure S5) shows that the mass estimated from the

SMPS, based on the assumptions of spherical particles and a particle mass density of 1.2 $g/cm^3$, reflects the dominant organic composition during that period. The comparison is strong over the 518 1-hour data points: $r^2 = 0.95$; slope = 0.90. As in the example in Figure S4, the $PM_{2.5}$ mass is estimated with the addition of the OPC data.

The evolution of PM concentrations in wildfire plumes is largely affected by the rate of secondary organic aerosol (SOA) formation (Akagi et al., 2012). Using the regression methods described in Wigder et al. (2013), $\Delta PM_1/\Delta CO$ is estimated at

0.09 (n=20, $r^2$=0.94) during the WHI1 to WHI2 period. This ratio is comparable to those of long-range transported plume (>740 km) of 0.03 and 0.06 found in Weiss-Penzias et al. (2007) and Wigder et al. (2013), respectively. Compared to the median ratios of 0.17, 0.29, 0.29, and 0.19, corresponding to distances of <140, 140-340, 340-540, 540-740 km, respectively, as reported by Wigder et al. (2013), the ratio found in this study is relatively low, suggesting that the net $PM_1$ loss exceeds SOA formation during transport as the plume ages. A recent wildfire study (Jolleys et al., 2012) found little SOA production

in aged biomass burning plumes; however, it was noted that other aircraft campaigns had found a large variability in OA enhancements.

The OC/EC ratio is also considered as its increase may indicate a contribution from SOA formation to the OC as a smoke plume ages. From the weekly-integrated $PM_1$ quartz filter samples, separated by day and night, the value of OC/EC was 3.8 (day) and 3.4 (night) at Whistler Peak for the period July 3-11. During the WHI1-WHI2 event, an OC/EC ratio of 3.7 was

estimated by scaling the week-averaged ACSM OM to the weekly-averaged filter OC and scaling the week-averaged rBC to the weekly-averaged filter EC. In the week prior to July $1^{st}$, 2012, a large variation in OC/EC (approximately 1.2 to 7) was observed in the suburb of Novosibrisk, approximately 400-800km south to southwest of the main sources of the Siberian wildfire (Smolyakov et al., 2014). Due to this large variation in OC/EC close to the source as attributed to variations in burning phase and fuel types, no further inferences of SOA formation in the plume were made. The ratios of m/z44 and

30  m/z43 to OM derived from the ACSM, indicate that the organic aerosol during the height of the BB plume (WHI1 to WHI2) was more oxygenated than at any other time during July 3-14 (Fig. S6). If SOA production during the plume transport was insignificant, then this observation implies either a relatively high level of oxygenation near the source, heterogeneous oxidation of the OM during plume transport, or the loss of less oxygenated components of the OM during transport.



The overall contribution of the July 6-10 Siberian Plume event to the air quality standards for $O_3$ and $PM_{2.5}$ at Whistler Peak were estimated using AURAMS baseline modelling (also shown in Fig. 6). Generally, the model depicts relatively clean air quality conditions (approximately 30 ppbv and 1.0 µg/m³ for $O_3$ and $PM_{2.5}$, respectively) on July 5th with an increase trend in baseline $O_3$ (between 5 to 10 ppbv from July 5-8). It is estimated that the Siberian wildfires contributed an additional 44

5    ppbv to the daily maximum 8-hr $O_3$ and 12 µg/m³ to the 24-hr $PM_{2.5}$ concentration (Table 4).

### 3.4 Lower Fraser Valley (LFV)

The UBC lidar data indicated a plume arrival time of 12:00 PST on July 6th (LFV1, Fig 8a) and a plume elevation of between 2 – 3.6 km ASL over Metro Vancouver which is similar to that found by the Whistler lidar but with generally weaker aerosol backscatter ratio readings (Fig. 8a). The plume had a shorter overall persistence over Metro Vancouver with the plume's

trailing edge shifting northward by 03:00 PST on July 8th (LFV2). It appears that the conditions favourable for the transport of Siberian smoke to LFV were also conducive to $O_3$ production (e.g. light winds, clear skies, and high temperatures). This makes the attribution of Siberian smoke impacts on local air quality slightly more challenging in the LFV than at Whistler. Figure 8 (b,c) depicts the increasing $O_3$ and $PM_{2.5}$ across the LFV during July 6-10 with the network average 1-hr $O_3$ and 1-hr $PM_{2.5}$ rises to 54 ppbv and 18 µg/m³, respectively, on July 8th. Peak $O_3$ condition of 79 ppbv observed at Hope Airport

(HA) and $PM_{2.5}$ of 26 µg/m³ at Port Moody Rocky Point (PMR, not listed in Table 3) arose as inland temperature reached a maximum of 31⁰C at Chilliwack (CA) on July 8th. Inland temperatures and $O_3$ conditions moderated subsequently on July 9th following a brief period of elevated convection in the early morning hours that affected parts of Washington State and LFV. High $O_3$ concentration were reported over the high elevation (>1000m ASL) site in North Vancouver by a separate field campaign (Bart et al., 2014).

The AURAMS model was used to assess the underlying baseline air quality (Fig. 8 b and c), and to assess the local impacts through a site-by-site analysis of differences with LFV monitoring network data (Fig. 9 a and b). For July 6-8, the AURAMS baseline simulation without wildfire emissions captured the rise in daytime $O_3$ on average across the LFV network (Fig. 8b) and on a site-by-site basis (Fig. 9a). In contrast, the baseline simulation was consistently lacking in nighttime $O_3$ titration. Previous air quality studies have shown that these discrepancies do not significantly affect the rest of the baseline AQ

simulation (Makar et al, 2014). As such, the estimates of $O_3$ enhancement within the LFV are based on daytime hours only. The 1-hr $O_3$ enhancements occurred on July 9-10 appears, based on the lidar data, to be about 18 - 27 hours after the plume advected north of the LFV (Fig. 9a). The average enhancement of peak 8-hr maximum $O_3$ across the LFV was estimated to be 10 ppbv (Table 4; variability shown in Fig. 9a). Spatial maps of 8-hr $O_3$ (Fig. S7) show widespread enhancements across the LFV on July 9th lingered in the western portion on July 10th. The lack of $O_3$ enhancement estimated throughout the LFV

on July 8th (Fig. 9a and Fig. S7) suggests the exceedance-level $O_3$ episode in Chilliwack (Fig. 9c) may have been due to local $O_3$ production and not the Siberian wildfire.

The $PM_{2.5}$ enhancements associated with the LRT event developed in the morning of July 8th, much earlier than with any $O_3$ enhancements, and then persisted through July 10th. The Siberian plume is estimated to have enhanced the 24-hr average





$PM_{2.5}$ by a maximum of 9 µg/m$^3$ for sites across the LFV. Spatially, the largest enhancements to $PM_{2.5}$ appear to be concentrated over the northern portion of the LFV (Fig. 4 and by day in Fig. S7). This suggests that entrainment of the plume may have been facilitated by slope flows along the northern mountainous terrain, yet appears not to have reached the floor of the LFV, as evidenced by $O_3$ data from the fixed monitoring network.

### 3.5 Washington State

In general, the air quality concentrations in Washington State (WA) during the episode were similar to those of the LFV. The average hourly $PM_{2.5}$ across WA rose to a maximum of ~15 µg/m$^3$ on July 9th and the peak 8-hr $O_3$ of 72 ppbv (Fig. S8) occurred at Enumclaw on July 8th. The LRT smoke enhancements in WA (Table 4) were similar to those in the

LFV for $O_3$ with a 10 ppbv increase to the 8-hr $O_3$ and a weaker contribution to the 24-hr $PM_{2.5}$ of 4 µg/m$^3$ (Fig. 4 and by day in Fig. S7). However, the timing and geographical extent of the impacts of $PM_{2.5}$ and $O_3$ differed considerably. In particular, the impacts on $PM_{2.5}$ were minimal over WA until July 10th, while $O_3$ underwent some modest enhancement on July 9th at a limited number of monitoring locations (Fig. S8). It is likely that some of the discrepancies (between the Canadian and US portions of the LFV in the estimated plume impacts) arise from model errors stemming from different

methodologies used to create the 2010 Canadian and 2008 US emissions inventories.

### 3.6 The Interior of British Columbia

The character of the Siberian plume and its impacts over the Interior of British Columbia differed considerably from the enhancements seen over the LFV and WA. The air quality for two communities (Fig. 10), Kamloops and Williams Lake, reflect the dramatic response seen across the Interior. The arrival of the plume at these communities, 19:00 PST on July 6

for Kamloops and 12:00 PST on July 8th for Williams Lake, resulted in a sudden shift in hourly $PM_{2.5}$ and elevated $O_3$ on the subsequent day. Based on the AURAMS modelling, it is estimated that the plume contributed an additional 18 and 37 ppbv at Kamloops and Williams Lake, respectively, to the daily maximum 8-hr $O_3$, and an additional 32 µg/m$^3$ and 30 µg/m$^3$, respectively, to the 24-hr $PM_{2.5}$ (Table 4). The suddenness and magnitude of these impacts suggests that the plume became well mixed into the boundary layer after its transition across the Coast Mountains, which also corroborates with the

entrainment signatures noted on the Whistler lidar that shows the plume mixed down to the base of the Whistler Valley (Fig. 5).

Unfortunately, the 12 km model study domain did not extend north sufficiently to provide baseline estimates of the air quality conditions to Quesnel, Prince George, and the Northern Interior, which also experienced conditions that exceeded national and provincial air quality standards. Instead, a baseline air quality level was inferred from the observations on July

6th and the enhancements at these sites (listed in Table 4) were found to be consistent with those experienced at Williams Lake (for a July 6th baseline) except for slightly weaker 24-hr $PM_{2.5}$ enhancements of 22-24 µg/m$^3$.



On July 10[th], as the bulk of the Siberian plume shifted northeastward out of British Columbia, remnants of the plume shifted over the Southern Interior and Whistler; however, attribution of the plume's impacts was impeded due to local wildfire activity (approx. 100 km west of Kelowna). An analysis of surface $PM_{2.5}$ concentration fields calculated from the GEM-MACH FireWork (Pavlovic et al., 2016) showed only localized smoke from wildfires in the Southern Interior on July 10[th] (Fig S9). Overall, smoke from the Siberian wildfires likely enhanced air quality impacts in this region by an additional 15 ppbv to the 8-hr O and 15 $\mu g/m^3$ to the 24-hr $PM_{2.5}$, based on the impact analysis conducted at the Vernon site (Fig. S10) for the July 6-9 period.

## 4. Conclusions

The extreme 2012 Russian boreal fire season resulted in the trans-Pacific transport of a large smoke plume across the Pacific in early July which had substantial, yet varied, impacts across large part of British Columbia and Washington State. Air quality concentrations in excess of the Canada Wide Standard (65 ppbv for 8-hr $O_3$) and the British Columbia Air Quality Objective (25 $\mu g/m^3$) of 24-hr $PM_{2.5}$ were experienced on July 6-10 at several municipalities across British Columbia.

The aged smoke plume (7-10 days old), as supported from data from Whistler Peak, had a mean normalized enhancement ratio ($\Delta O_3 / \Delta CO$) of 0.26 and a $PM_1/CO$ ratio of 0.09, consistent with other long-range transport smoke studies of similar origin. Fine organic aerosol mass (~87% based on the ACSM measurements) accounted for the majority of the additional 1-hr and 24-hr $PM_{2.5}$ enhancements of 28 $\mu g/m^3$ and 14 $\mu g/m^3$, respectively. Analysis of OC/EC did not provide evidence for significant SOA production during transport as the plume ages. However, analysis of the ratios of m/z44 and m/z43 to OM derived from the ACSM, the level of oxygenation of organic material in this plume was much higher than during flanking times. If SOA formation during transport was minimal, then the relatively high level of oxygenation may have resulted from three factors: the OM near the source was relatively oxygenated; heterogeneous oxidation of the OM during plume transport; or removal of less oxygenated components of the OM during transport.

Baseline air quality modelling provided by the AURAMS model was used to delineate the scope of the LRT enhancements to air quality under varying atmospheric conditions. Generally, the largest enhancements to 8-hr $O_3$ [34-44 ppbv] and 24-hr $PM_{2.5}$ [14-32 $\mu g/m^3$] occurred at Whistler Peak, Kamloops and across the Central and Northern Interior of British Columbia. In contrast, lesser LRT enhancements of [10-12 ppbv] 8-hr $O_3$ and [4-9 $\mu g/m^3$] 24-hr $PM_{2.5}$ were estimated for coastal British Columbia and Washington State. Stable atmospheric conditions along coastal area of British Columbia and Washington State likely limited the initial entrainment of the plume. The AURAMS modelling approach accounted for enhancements in anthropogenic air quality due to the smoke plume in the majority of cases with the exception of July 8[th] at Chilliwack and Enumclaw, where elevated levels may have been due to other factors.

Long range transport smoke events, such as the 2012 Siberian forest fire plume, highlight the far reaching impacts of wildfires and how air quality impacts can vary with downwind meteorology, plume age and entrainment pathways. This demonstrates the need to include wildfire emissions within chemical transport model and to derive effective



parameterizations for the lofting and subsequent reactions in modelling ambient air quality conditions. Wildfires sources pose an increasing concern as global climate change is expected to increase wildfire frequency and severity in many regions (IPCC, 2007; Gillett, 2004; Liu et al., 2013). As such, wildfire smoke is expected to contribute more to the ambient levels for $O_3$, $PM_{2.5}$, CO, and other pollutants and can potentially increase the frequency of extended periods of degraded air quality.

**Data availability**

Data sources used in this paper are listed in the supplemental material (Table S1). For ECCC data not readily accessible, please contact the corresponding author (Andrew.Teakles@canada.ca) for assistance.

**Supplementary material related to this article is available online at:**

**Author contribution**

A. Teakles designed the study, developed the methodology and prepared the manuscript with contributions from co-authors.

R. So consolidated and analyzed the data and provided contributions to the methodology.

J. Baik provided assistance in developing R code to generate figures.

B. Ainslie generated the AURAMS model run.

S. Hanna, A. Bertram and L. Huang provided data and their expertise on data and instrumentation comparability.

K. Strawbridge provided the lidar data and his expertise in interpreting this data.

AM. Macdonald and R. Leaitch provided and analyzed data collected at Whistler Peak High Elevation site.

D. Toom assisted with the ACSM data and supplementation analysis on OA oxygenation.

I. McKendry provided his expertise on the LFV analysis.

R. Vingarzan reviewed and provided contributions to the methodology and manuscript composition.

C. Schiller reviewed the manuscript and provided her chemistry expertise to the data interpretation.

R. Nissen performed the synoptic analysis.

D. Jaffe contributed to the enhancement ratios analysis and the U.S. analysis.

*Acknowledgements. We thank Jennifer Chudak and Sharon Gunter from the BC Ministry of Environment for providing the $O_3$ and $PM_{2.5}$ data for the Lower Fraser Valley and to Natalie Suzuki for air quality climatology information. We would like to acknowledge Environment and Climate Change Canada (ECCC) and, the provincial, territorial and regional governments as partners of the National Air Pollution Surveillance (NAPS) Program for the use of the time integrated ambient air quality data. We thank Dr. Ewa Dabek-Zlotorzynska of Analysis and Air Quality Section, ECCC for providing*





*additional NAPS air quality 24-hr reconstructed PM and levoglucosan dataset that were not included in this study but were helpful in our initial assessments. We acknowledge the use of Rapid Response imagery from the Land, Atmosphere Near real-time Capability for EOS (LANCE) system operated by the NASA/GSFC/Earth Science Data and Information System (ESDIS) with funding provided by NASA/HQ. Thanks to Esther Fan for her early assessment of the Siberian Plume within the Lower Fraser Valley and to Dr Paul Makar for his advice on numerical modelling. Finally, we would like to thank Ryan Mason, Jack Chen, and Paul Makar for their useful discussion regarding this manuscript.*

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



**Table 1. Instrument list for various monitoring networks used in the study**

| Network/ Site | Measured Parameters | Instrument(s) | Reference |
|---|---|---|---|
| BCMoE | $O_3$ | Thermo Environmental Instruments Inc., UV absorption monitor (TECO 49C) | Environment Canada (2013) |
| | $PM_{2.5}$ | Thermo Scientific TEOM 1400ab with sample equilibration system | |
| | | Met One BAM-1020 | |
| | | Thermo Scientific Model 5030 SHARP | |
| Ucluelet | $O_3$ | Thermo 49i $O_3$ analyzer | McKendry et al. (2014) |
| | $PM_{2.5}$ | Thermo Sharp 5030 particulate monitor | |
| WSMN | $O_3$ | Teledyne-API T400 | WA ECY (2008 & 2012) |
| | $PM_{2.5}$ | Nephelometer | |
| Cheeka Peak | $O_3$ | Teledyne-API T400 | WA ECY (2008 & 2012) |
| | $PM_{2.5}$ | Radiance Research M903 | |
| Mt. Rainier Jackson Visitor Center | $O_3$ | Teledyne-API 400 | WA ECY (2008 & 2012) |
| CORALNet | aerosol backscatter | Lidar | Strawbridge (2013) |
| Whistler Peak High Elevation site | CO | Thermo Environmental Instruments Inc., Model 48C-Trace Level analyzer | Macdonald et al. (2011) |
| | $O_3$ | Thermo Environmental Instruments Inc., UV absorption monitor (TECO 49C) | Macdonald et al. (2011) |
| | Refractory Black Carbon | Droplet Measurement Technologies Inc., Single Particle Soot Photometer (SP2) | Takahama et al. (2011) |
| | Particle Chemistry < 1 micron ($SO_4^{2-}$, $NO_3^-$, $Cl^-$, $NH_4^+$, Organics) | Aerodyne Aerosol Chemical Speciation Monitor | Takahama et al. (2011) |
| | Particle size distributions from 0.25 to 32 µm diameter | GRIMM portable aerosol spectrometer 1.109 | Grimm Technologies Inc. (2010) |
| | Particle size distributions in range ~ 0.01 nm to 0.50 nm diameter | TSI 3081 Scanning Mobility Particle Sizer (SMPS) with a TSI 3775 Condensation Particle Counter (CPC) operated at a low flow setting | McKendry et al. (2008) |





**Table 2: 2012 Ambient air quality standards and objective for $O_3$ and $PM_{2.5}$ used in the study**

| Standard/ Objective | Pollutant | Averaging period | Value |
|---|---|---|---|
| Canada NAAQS[a] | $O_3$ | 1-hour | 82 ppb |
| Canada CWS[b] | | | 65 ppb |
| BC provincial and Metro Vancouver[c, d] | $O_3$ | 8-hour | 65 ppb |
| U.S. EPA NAAQS[e] | | | 75 ppb |
| Canada CWS[b] | | | 30 µg/m$^3$ |
| BC provincial and Metro Vancouver[c, d] | $PM_{2.5}$ | 24-hour | 25 µg/m$^3$ |
| U.S. EPA NAAQS[e] | | | 35 µg/m$^3$ |

[a] Health Canada (1999)

[b] CCME (2014)

[c] BC MOE (2014)

5   [d] Metro Vancouver (2014)

[e] U.S. EPA (2014)





**Table 3. Summary of ambient $O_3$ and $PM_{2.5}$ conditions for stations that had severe (in bold) air quality degradation (exceeding one of the values listed in Table 2) between July 6th and July 11th. Values indicated reflect the maxima of 1, 8, and 24-hr averaging periods. Values in bracket denote historical July average daily maxima of 1 and 8-hr average periods and the historical July average of 24-hr running mean.**

| Region | Station(s) | Station(s) ID | $O_3$ (ppbv) | | $PM_{2.5}$ ($\mu g/m^3$) | |
| --- | --- | --- | --- | --- | --- | --- |
| | | | 8-hr | 1-hr | 24-hr | 1-hr |
| LFV | Chilliwack Airport | CA | **67 (35)** | 74 (41) | 14 (6) | 21 (12) |
| | Hope Airport[c] | HA | 64.8 (39) | 79 (43) | 18 (6) | 21 (11) |
| Whistler | Whistler Peak High Elevation Site | WHI | **83** | **86** | 12 | 24 |
| Southern | Kelowna College | KC | **74 (40)** | 78 (44) | 15 (5) | 30 (13) |
| Interior | Kamloops Fire Station | KFS | **65 (39)** | 72 (43) | **36 (5)** | 44 (12) |
| | Vernon Science Centre | VSC | **67 (32)** | 73 (36) | 19 (5) | 30 (12) |
| Central | Williams Lake[a] | WL* | **80 (34)** | **84 (37)** | **34 (5)** | 46 (16) |
| Interior | Quesnel[a] | QL* | **76 (31)** | **92 (35)** | **31 (6)** | 57 (20) |
| | Prince George[a] | PG* | NA | NA | **31 (6)** | 49 (17) |
| Northern | Burns Lake Fire Centre | BL | NA | NA | **28 (3)** | 38 (9) |
| Interior | Houston Firehall | HF | NA | NA | **27 (3)** | 38 (9) |
| (N. INT) | Telkwa | TK | NA | NA | **26 (3)** | 36 (11) |
| U.S.[b] | Enumclaw | ENC | **72** | 80 | NA | NA |

5  [a] multiple stations in the area of interest

[b] encompasses coastal stations in northwestern Washington State as shown in Figure 1

[c] air quality conditions at Hope Airport are noted here as it nearly exceeded the regional 8-hr $O_3$ objective




**Table 4. Maximum daily enhancement (observed – model) of daily maximum 8-hr rolling average O$_3$ and 24-hr rolling average PM$_{2.5}$ based on various air quality baselines (4km and 12km AURAMS and historical values) from July 6$^{th}$ to July 11$^{th}$, 2012**

| Area of Interest | Baseline | O$_3$ (ppbv) | | | PM$_{2.5}$ ($\mu$g/m$^3$) | | |
| --- | --- | --- | --- | --- | --- | --- | --- |
| | | No. of Sites | enhancement | uncertainty range | No. of Sites | enhancement | uncertainty range |
| Whistler Peak | AURAMS (4km) | 1 | 44 | (16, 49) | 1 | 12 | (8, 12) |
| LFV[a,b] | | 19 | 10 | (2, 23) | 17 | 9 | (6, 13) |
| Vancouver Island | | 6 | 14 | (2, 27) | 9 | 9 | (7, 13) |
| Ucluelet | | 1 | 12 | (0, 15) | 1 | 7 | (1, 6) |
| U.S. [a,b,c] | | 10 | 10 | (0, 25) | 10 | 4 | (4, 12) |
| Williams Lake | AURAMS (12km) | 1 | 37 | (20, 57) | 2 | 30 | (24, 31) |
| Kamloops | | 1 | 18 | (22, 33) | 1 | 32 | (29, 33) |
| Vernon | | 1 | 15 | (17, 33) | 1 | 15 | (10, 13) |
| Kelowna | | 1 | 6 | (5, 20) | 1 | 8 | (10, 11) |
| Northern Interior | Historical value (Jul 6th) | 1 | 34 | - | 4 | 22 | - |
| Prince George | | NA | NA | | 2 | 23 | |
| Quesnel | | 1 | 43 | | 3 | 24 | |
| Williams Lake | | 1 | 38 | | 2 | 31 | |

[a] estimated enhancement excluding July 10$^{th}$ and July 11$^{th}$

[b] O$_3$ enhancements were based on daytime values

5   [c] encompass northwest coastal stations in Washington State as shown in Figure 1



**Table S1. Source information and data availability**

| Site | Data | Source |
|---|---|---|
| British Columbia surface monitoring | O₃, PM₂.₅ | BCMoE (http://envistaweb.env.gov.bc.ca/) |
| Lower Fraser Valley (LFV) AQ sites | O₃, PM₂.₅ | BCMoE (http://envistaweb.env.gov.bc.ca/) |
| BC surface and LFV AQ sites | O₃, PM₂.₅ (climatology) | ECCC NAPS website (annual summaries: http://maps-cartes.ec.gc.ca/rnspa-naps/data.aspx) |
| Ucluelet MBL site | O₃, PM₂.₅ | ECCC ,contact info: Andrew.Teakles@canada.ca |
| Whistler Peak High Elevation Site | Particle size distribution , O₃, CO, Filter packs IC (tSO4), Black Carbon, TOT EC & OC, ACSM, lidar | ECCC, contact info: Andrew.Teakles@canada.ca |
| University of British Columbia | Lidar | ECCC, contact info: Andrew.Teakles@canada.ca |
| U.S. sites (Anacortes, Custer, Enumclaw, Issaquah, Lacey College St, Mount Vernon, Mt. Rainier Jackson Visitor Center, Seattle Beacon Hill, Yelm) | O₃, PM₂.₅ (neph) | Puget Sound Clean Air website (http://airgraphing.pscleanair.org/) |
| U.S. sites (Bellingham, Queen Anne, North Bend, Lynnwood, Duwamish) | O₃, PM₂.₅ (neph) | Puget Sound Clean Air website (http://airgraphing.pscleanair.org/) |
| U.S. Background site - Mt. Bachelor | O₃ | For final QC'd, contact Dr. Dan Jaffe at djaffe@u.washington.edu |
| U.S. Background site - Cheeka Peak | O₃, PM₂.₅ | Department of Ecology, State of Washington, https://fortress.wa.gov/ecy/enviwa/ |
| All sites | AURAMS O₃, PM₂.₅ | ECCC, contact info: Andrew.Teakles@canada.ca |
| World | MODIS AOD | http://modis.gsfc.nasa.gov/data/ |
| Pacific Northwest & Eastern Russia | MODIS true colour | http://visibleearth.nasa.gov/view.php?id=78406 |
| Quillayute & Kelowna | Radiosondes | http://weather.uwyo.edu/upperair/sounding.html |
| Elevation data (for fig 1) | GLOBE Elevation | http://www.ngdc.noaa.gov/mgg/topo/globe.html |
| Canada/US | GEM-MACH FireWork | ECCC, contact info: Andrew.Teakles@canada.ca |





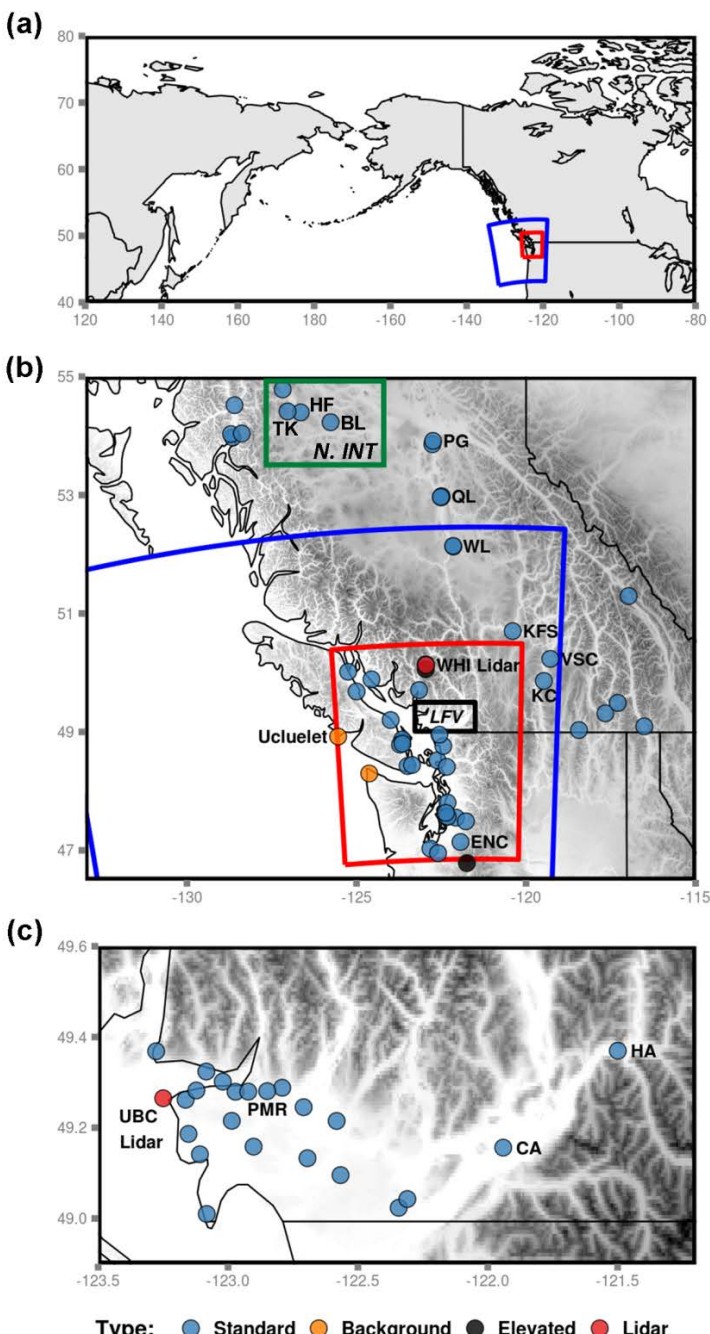

**Fig. 1.** Map of the monitoring network data and AURAMS modelling domain used in the study. The blue and red outlined areas in both (a) and (b) highlight the 12 km and 4 km AURAMS grid domains, respectively. The green outline in (b) highlights the Northern Interior sites that fall outside the gridded domain. The Lower Fraser Valley inset (c) displays the Metro Vancouver air quality monitoring network and the location of the UBC lidar. Abbreviations used in this figure are defined in Table 3 and throughout the text. Topography shaded in panels (b) and (c).





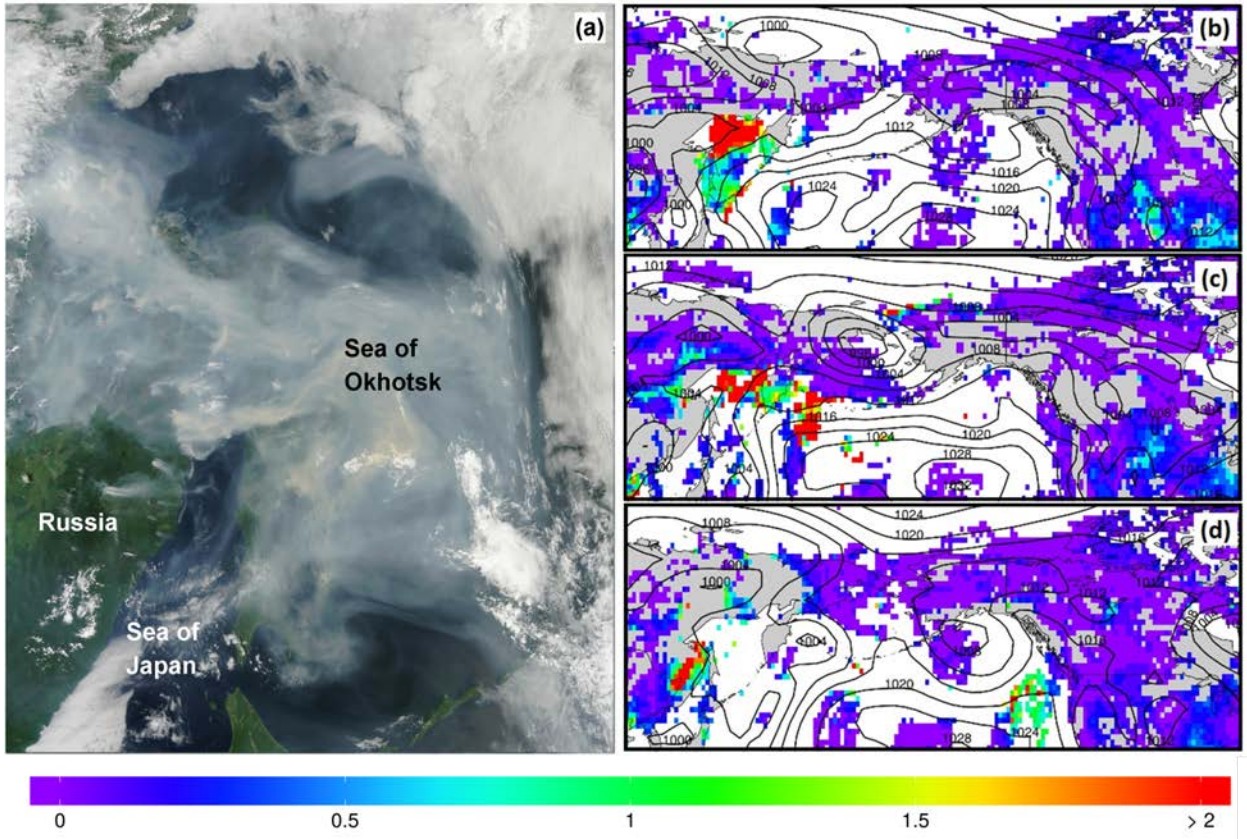

**Fig. 2. MODIS true colour Image for June 29th (a) from the NASA Earth Observatory site showing the buildup of smoke from Eastern Russia drifting across the Sea of Japan, Sea of Okhotsk and Kamchatka peninsula (east of Sea of Okhotsk). All MODIS true colour images provided courtesy of Jeff Schmaltz and the MODIS Land Rapid Response Team, NASA GSFC. The daily average MODIS Aerosol Optical Depth product for July 1st (b) and 3rd (c) illustrates the eastward progression of the Siberian Fire plume in early July onto the Western Pacific prior to its arrival off the coast of Washington and Oregon States on July 6th, 2012 (d). The solid black line in panels b, c, and d indicate the mean sea level pressure contours (mb).**





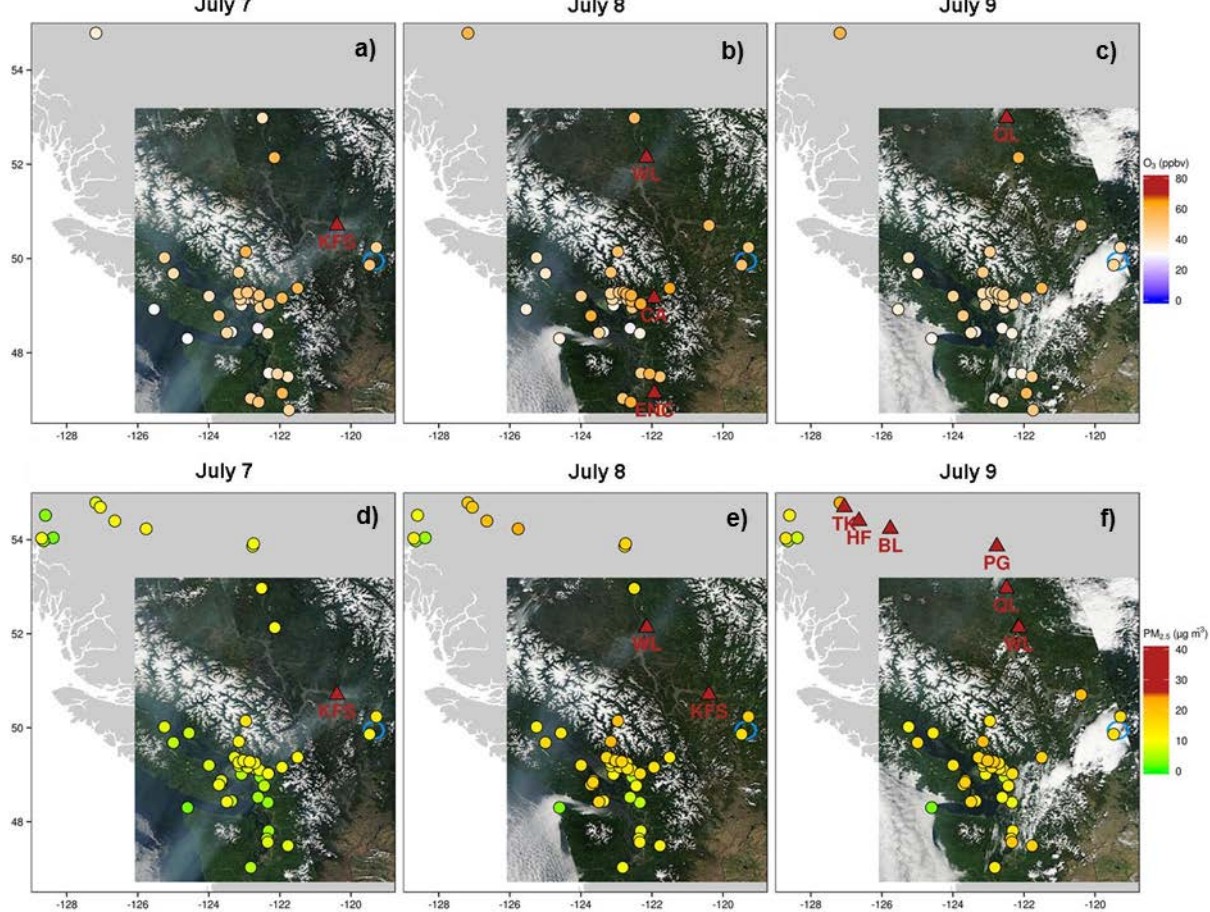

**Fig. 3. MODIS true colour Image composite maps for July 7, 8, and 9, 2012 overlaid with daily maximum 8-hr O₃ (a, b, and c, respectively) and 24-hr PM₂.₅ (d, e, f, respectively) for the study sites. The locations where observed concentration exceed the national or regional air quality standards for either O₃ or PM₂.₅ are indicated in red triangles in the appropriate panel.**





**Fig. 4. Overview of the maximal enhancements to 8-hr O$_3$ and 24-hr PM$_{2.5}$ estimated for the study area (a and b, respectively) and the LFV (c and d, respectively) based on the differences between the ambient air quality observation and the AURAMS baseline modelling for the Siberian smoke plume event.**




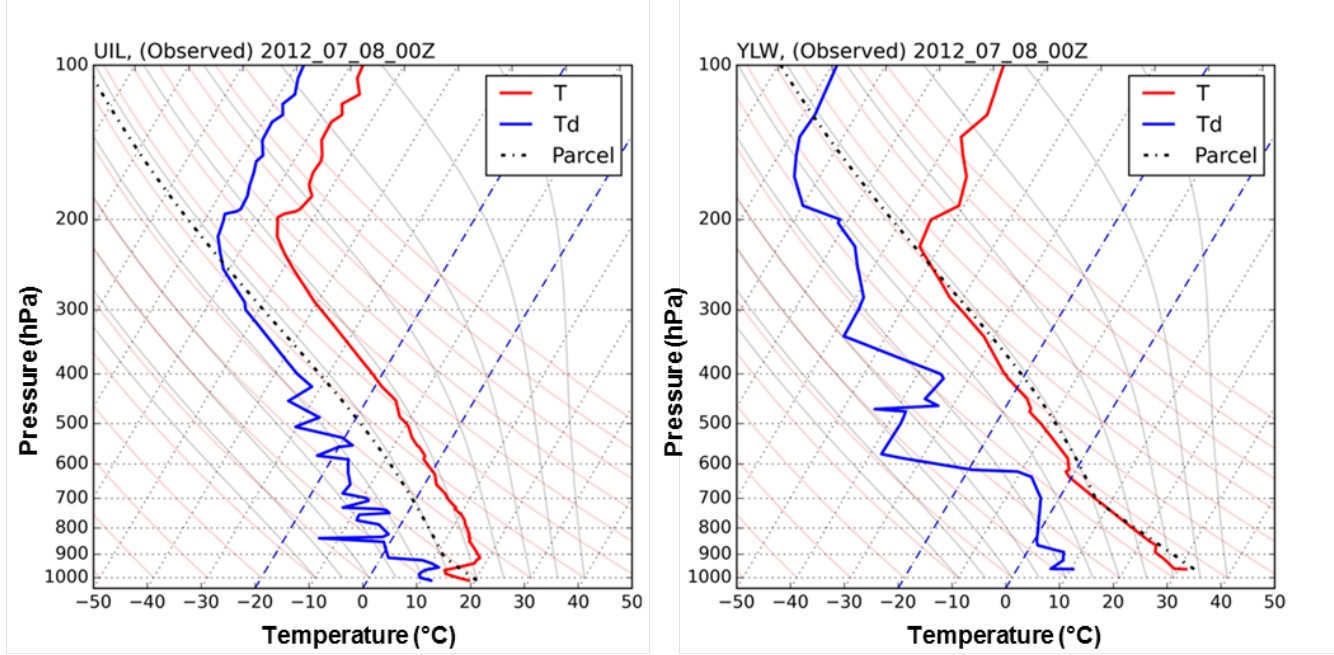

**Fig. 5. Skew-T atmospheric radiosondes for Quillayute (UIL) and Kelowna (YLW) at 00Z on July 8[th], 2012. The atmospheric dry bulb (T) and dewpoint temperature (Td) are shown from the surface up to 100 mb. Black dashed line is a trace of a surface parcel lift.**



**Fig. 6. Whistler lidar (650 m ASL) measured backscatter ratio of the 1064 nm channel to that of clear air for the July 2012 Siberian smoke plume (a) and observed (solid) 1-hr O₃ (b), 1-hr CO (c), and 1-hr fine particulate matter (d) at Whistler Peak High Elevation station (2182 m ASL, indicated as a horizontal blue line in panel a). AURAMS baseline (dashed) 1-hr O₃ and 1-hr PM₂.₅ are shown in panel b) and d), respectively. The shaded regions indicate night time hours between 18:00 PST to 07:00 PST.**





**Fig. 7. Aerosol chemistry measurements taken at Whistler Peak High Elevation site between July 6th 2012 and July 12th, 2012. Hourly Organics(Org), NO$_3^-$, NH$_4^+$, SO$_4^{2-}$ measured by the ACSM (a, b), EC and OC as sampled using the EnCan Total-900 thermal method (TOT) (a), particle SO4 (tSO4) as sampled by quartz filter pack (FP) (b), and rBC as acquired using the Single Particle Soot Photometer (SP2) (c). The shaded regions indicate night time hours between 18:00 PST to 07:00 PST.**



**Fig. 8. UBC lidar observations and air quality conditions across the Lower Fraser Valley network between July 6th and 12th, 2012. Panel (a) shows the ratio of measured backscatter of the 1064 nm channel to that of clear air. Panel (b) and (c) indicates the mean observed (solid) and AURAMS baseline without wildfire emissions (dashed) 1-hr O$_3$ (ppbv) and 1-hr PM$_{2.5}$ (µg/m³) for 19 and 17 sites, respectively, across the LFV air quality monitoring network. The light and dark grey shading represents the range and inter-quartile range, respectively, across the network at a particular hour. The vertical shaded regions indicate night time hours between 18:00 PST to 07:00 PST.**





**Fig. 9.** Contributions of the Siberian wildfire smoke plume based on the differences of observed 1-hr $O_3$ (a) and $PM_{2.5}$ (b) and the AURAMS baseline without wildfire emissions on a site-by-site basis for the LFV. For (a) and (b), the average contribution is labelled by the blue solid line. The light grey and dark grey shading represents the range and inter-quartile range of the contribution across the network at a particular hour. Panel (c) shows the AURAMS baseline and observed $O_3$ timeseries for Chilliwack, BC. The light blue shaded regions indicate night time hours between 18:00 PST to 07:00 PST





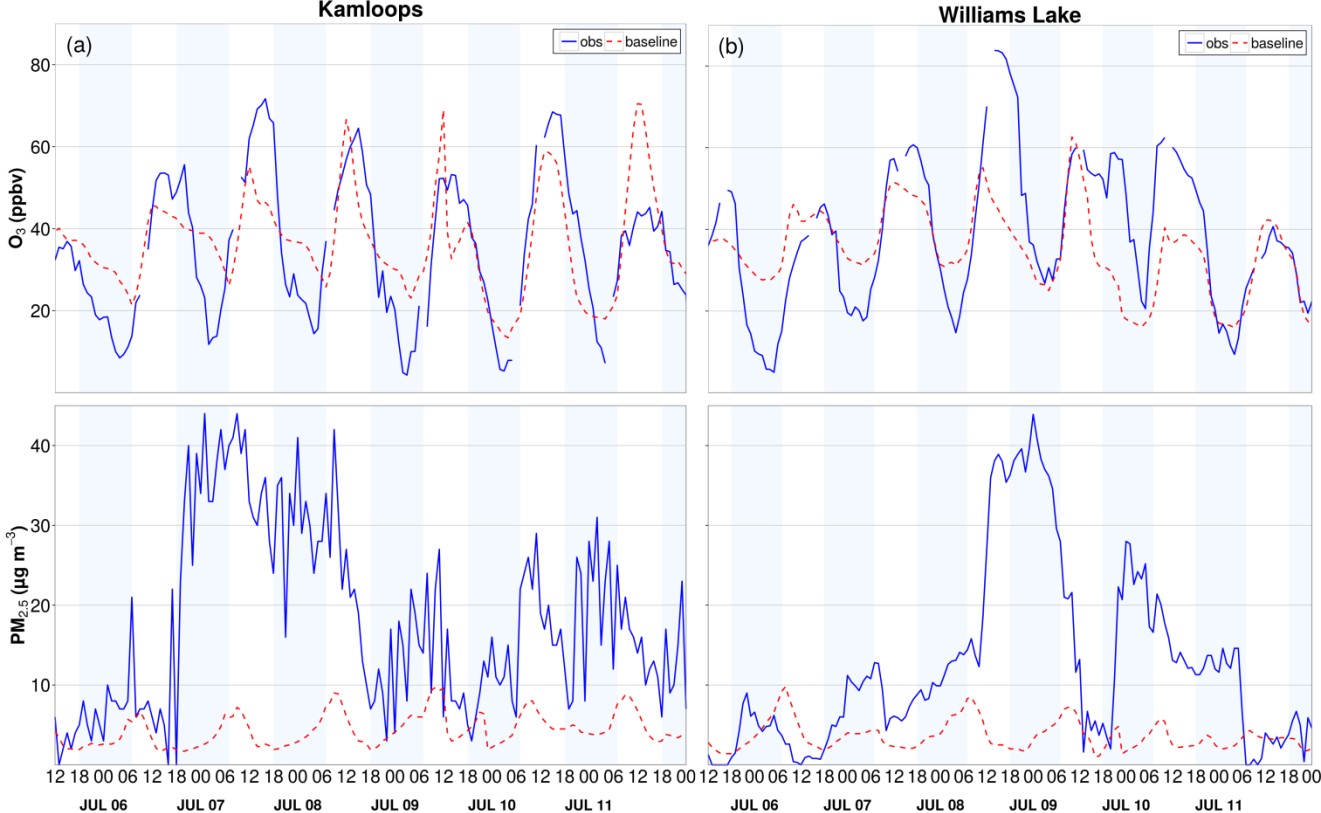

**Fig. 10.** The hourly observed (blue) and baseline AURAMS (red, dashed) without wildfire emissions O$_3$ (top) and PM$_{2.5}$ (bottom) for Kamloops (a) and Williams Lake (b). The shaded regions indicate night time hours between 18:00 PST to 07:00 PST.




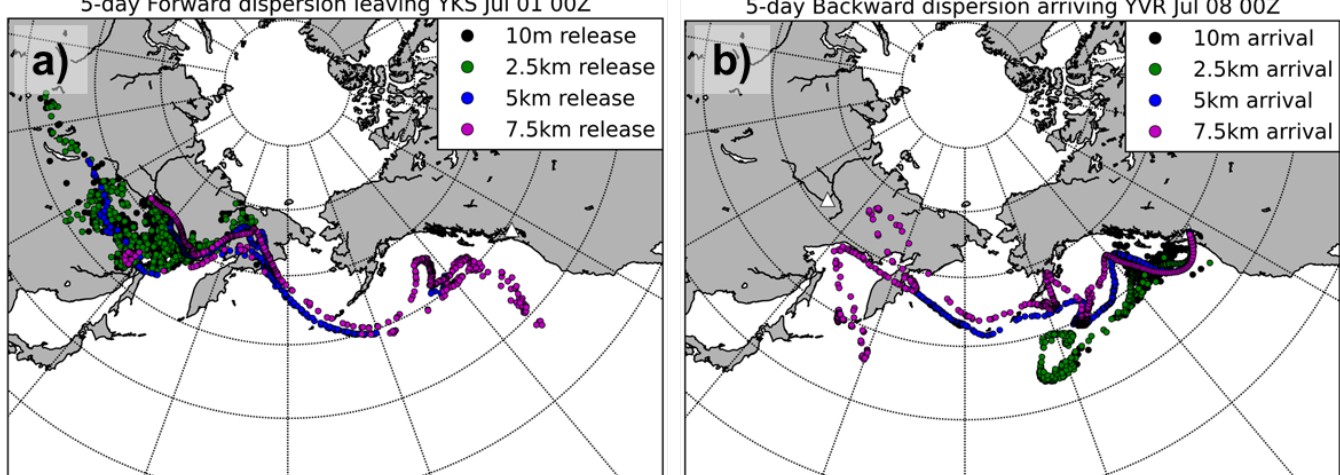

**Fig. S1.** **Five day HYSPLIT particle dispersion modelling from Yakutsk, Russia (YKS) released on July 1ˢᵗ, 2012 at 00Z (a) and to Vancouver, Canada (YVR) arriving at July 8ᵗʰ, 2012 at 00Z (b) with particle release heights of particle release heights of 10 m, 2.5 km, 5 km, 7.5 km AGL.**



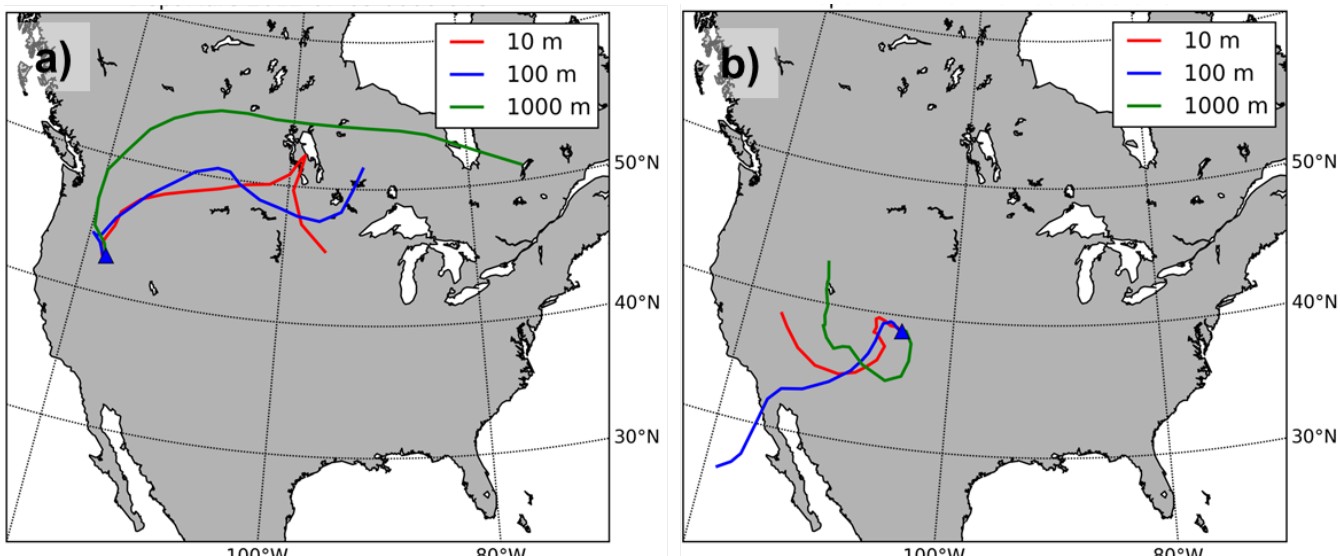

**Fig. S2. Five day forward trajectories using the CMC Trajectory model from the Long Draw, Oregon (a) and Waldo Canyon, Colorado wildfires (b) with release heights of 10 m, 100 m, and 1 km AGL.**



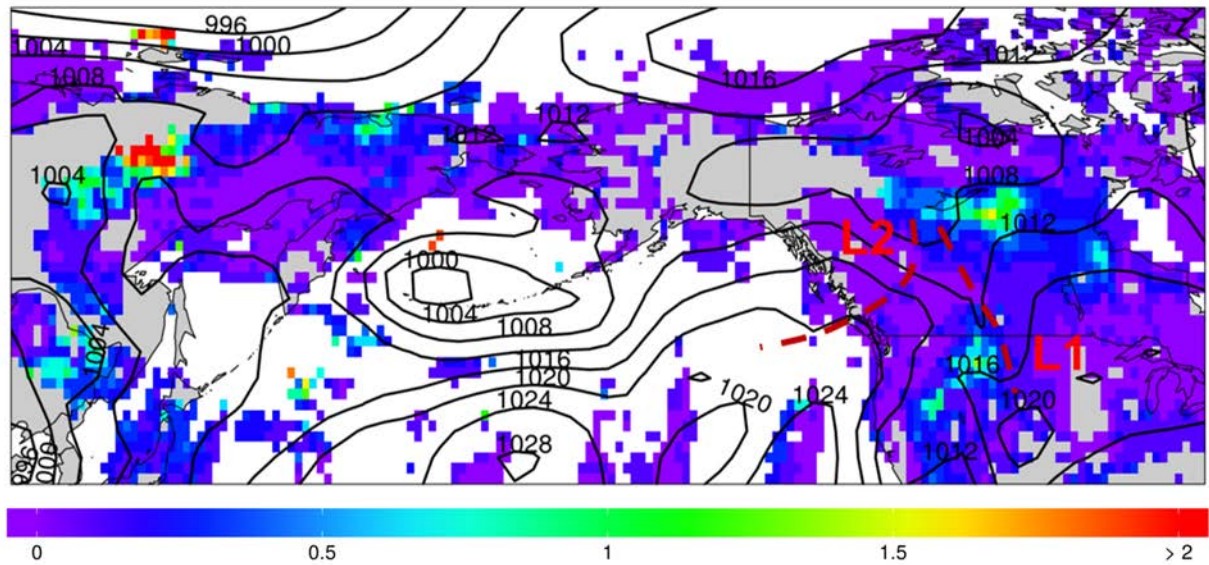

**Fig. S3. The daily average MODIS Aerosol Optical Depth product on July 10th 2012 showing the majority of the enhanced AOD values have shifted eastward out of the Pacific Northwest domain. L1 and L2 indicate troughs (dashed red) in the MSLP contours (solid black) associated with the advection of the plume across the Interior of British Columbia.**





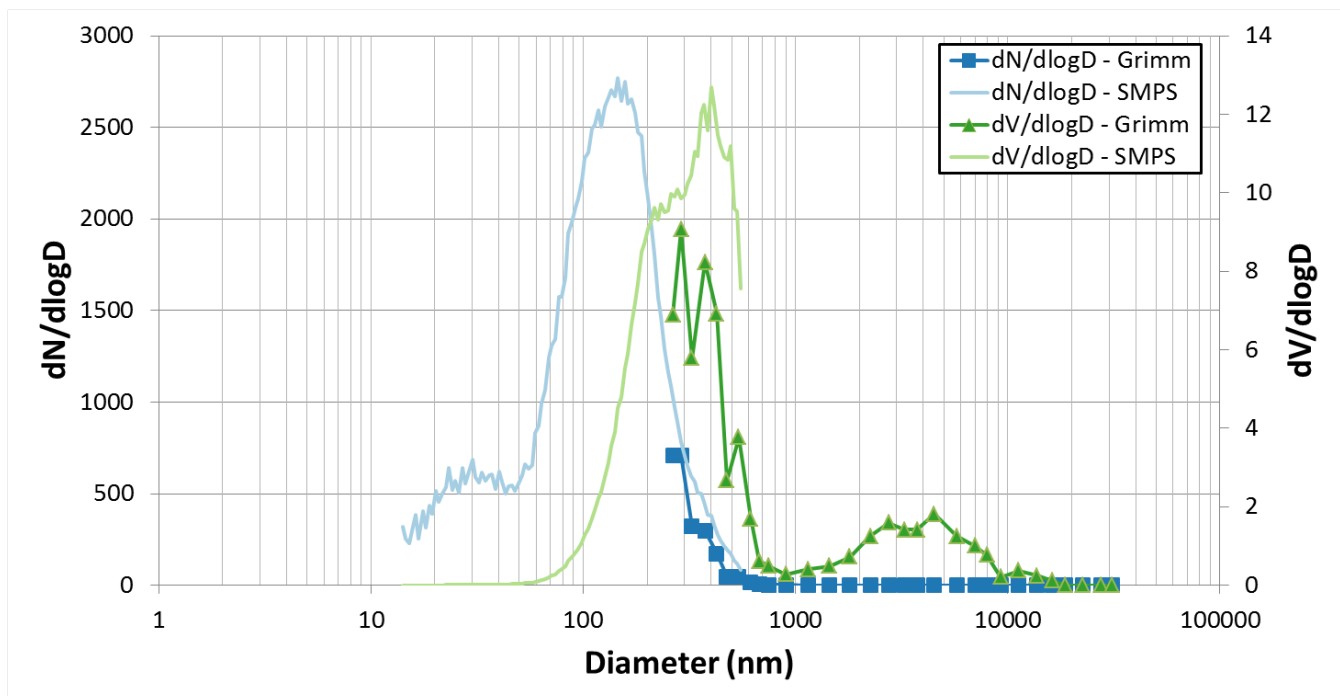

**Fig. S4. Comparison of SMPS and GRIMM aerosol number and volume distributions for July 9th, 2012 16h30 -20h30 PST.**





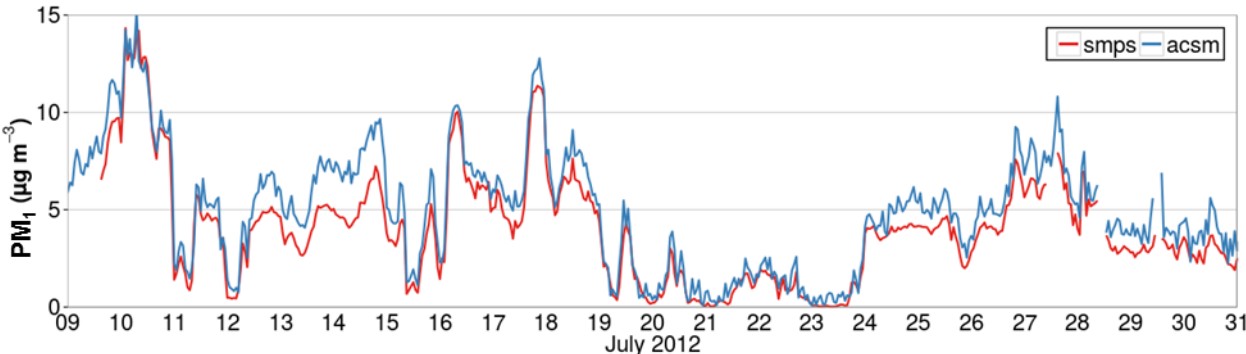

**Fig. S5.** Timeseries of SMPS PM$_1$ and ACSM readings between July 9$^{th}$ and July 31$^{st}$, 2012. The ACSM data have been adjusted for a collection efficiency of 0.5.


**Fig. S6.** The m/z44 and m/z43 components of the organic aerosol spectrum derived from the ACSM spectra. The ratio of organic signal at m/z44 (Org43) vs. m/z43 (Org44) is compared in a) to show that the organic aerosol was more oxygenated during the LRT smoke event (July 6[th] 14:00 PST to July 8[th] 06:00 PST) than at any other times during the 12 day period from July 3 to 14, 2012 (ACSM organic aerosol mass described in b).





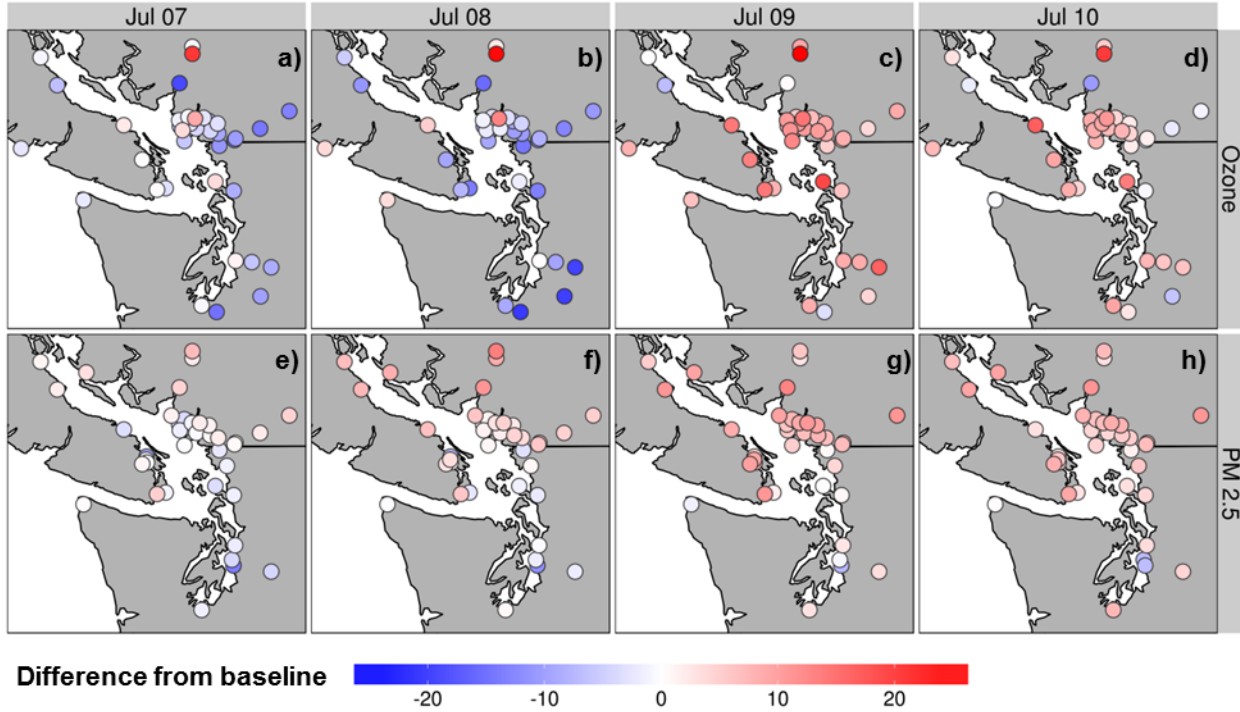

**Fig. S7. Spatial map of the differences of the AURAMS baseline without wildfire emissions to daytime averaged 8-hr O$_3$ (ppbv, panels a, b, c, d) and of the daily averaged PM$_{2.5}$ (μg/m$^3$, e, f, g, h) from July 7$^{th}$, 2012 to July 10$^{th}$, 2012.**




**Fig. S8.** Hourly average observed (solid) and AURAMS baseline (dashed) without wildfire emissions: hourly $O_3$ (a) and $PM_{2.5}$ (b) for sites across the Washington State air quality monitoring network. The light grey and dark grey shading represents the range and inter-quartile range (IQR) across the network at a particular hour. The shaded regions indicate night time hours between 18:00 PST to 07:00 PST.





**Fig. S9. Forecast North American wildfire emissions contribution to the total PM concentration (μg/m³) at the surface from the experimental GEM-MACH FireWork modeling system for: 18 UTC July 9th, 2012 (a), 10 UTC July 10th, 2012 (b), 18 UTC July 10th, 2012 (c), 18 UTC July 11th, 2012 (d).**





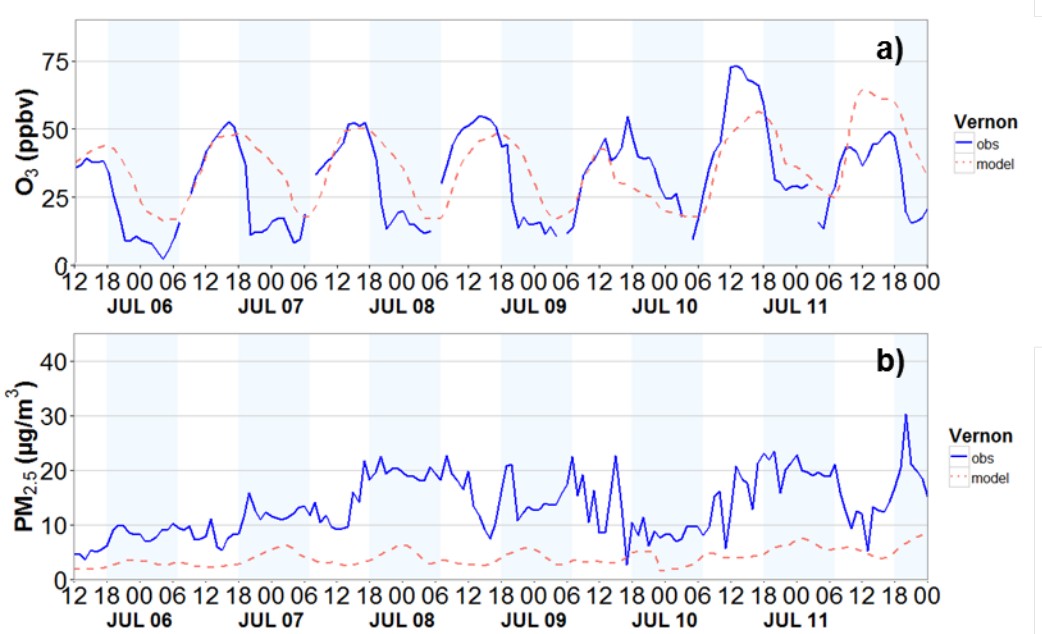

**Fig. S10. The observed and modelled baseline hourly O$_3$ (a) and PM$_{2.5}$ (b) for Vernon. The shaded regions indicate night time hours between 18:00 PST to 07:00 PST.**