# Peer review of "Impacts of the July 2012 Siberian Fire Plume on Air Quality in the Pacific Northwest"

_Atmospheric Chemistry and Physics, 2016_

## Referee Comment (RC1) · Anonymous Referee #2 · 5 Aug 2016

Acp-2016-302 "Impacts of the July 2012 Siberian Fire Plume on Air Quality in the Pacific Northwest" Teakles et al. Review:

I find this manuscript to be reasonably well written, though the amount of material it is attempting to cover is large, causing a general lack of focus. This manuscript attempts to cover a wide ranging of concepts, including biomass burning plume long range transport observations, boundary layer entrainment processes, air quality conditions over a relatively wide geographical area, air quality exceedances for several Canadian and US standards, biomass burning plume chemical analysis, and an assessment of the enhancements in several air quality measures – using an atmospheric model without biomass burning plume as the baseline. In some senses, this work ties these elements together knowing that this is what it might take to provide fuller understanding and predictive capabilities of the air quality effects of biomass burning events. However, given

the amount of material covered and the issues inherent with organization of this material, I feel that it dilutes any overall scientific significance of this particular manuscript. From my perspective, the most informative and useful portion of the manuscript is well summarized in the first two paragraphs of the conclusions. Here, the authors focus on the geographic range impacted by a specific long range transported biomass burning plume, the observed air quality exceedances, observations of the plume chemical characteristics, and comparisons with a baseline model run in an attempt to assess the quantitative impact of the plume on air quality. I expand on these comments below with some suggestions to try and focus the manuscript some and make it more significant.

The topic is timely and appropriate for ACP.

With appropriate attention paid to comments from reviewers, this manuscript should be published.

Major comments:

1.) This work appears to be a direct continuation of the Cottle et al. 2014 Atm Env paper on the same topic, but reprises a lot of what is already in Cottle et al.. For example, there are essentially three Figures that were taken, in part, from Cottle et al. and reproduced in this manuscript. Section 3.1 describes these figures to discuss the transport of the biomass burning across the Pacific. What is not detailed is what is new here on this topic. Perhaps the estimate of >6 days in transport? The authors note that they have done additional back trajectory studies, but it is not clear why. Perhaps due to greater geographical area under study? Do any of these issues make or break the interpretation/conclusions?

It would make the manuscript more manageable if this portion of the manuscript were removed and the manuscript focused on the geographic range, air quality exceedances, chemical characteristics, and assessment of quantitative impact.

For example, section 3.1 could be merged into the introduction in some manner, relying

on Cottle et al. to describe the Siberian fires, the transport of plumes across the Pacific, and the general region in Canada/US of impact, in detail. Figure 2 could be removed or placed in SI as it duplicates a portion of a Figure from Cottle and focuses only on transport across the Pacific.

2.) The results and discussion section of the manuscript is organized more or less along geographic lines of specific air quality networks (i.e., Whistler, Lower Fraser Valley, etc.), but each section has different information discussed therein, based on the measurements available at a given site/network. The section 3.2 "July 6-10 smoke event overview" contains most, but not all of the discussion of the geographic extend and boundary layer entrainment. This organization leads to Figure 3 connecting satellite observations to time and geography, Figure 4 showing overview of the enhancements of air quality observations compared with modeled baseline conditions which are discussed in more detail later, Figure 5 showing entrainment conditions, more related to Fig. 3 than Fig. 4, all followed by sections on geographic locations and more details on the air quality modeling results. This puts the detailed discussions of plume chemistry and quantitative assessment of air quality parameter enhancements at the end of the figures and mixed in with geographic details.

Furthermore, while the data was collected for specific networks of sites (i.e., Washington State in US, LFV, Whistler, and Interior), the discussion of the results suggest that the real impact was limited on coastal areas (i.e., parts of Washington State and LFV) and more significant on the in-land sites (interior and Whistler). Thus, the discussion/organization could focus more on the geographic effects rather than the specific network site locations.

Another potential organization might setup sections focusing on (a) plume impacts by geography (i.e., time and space), boundary layer entrainment, and air quality observations, (b) plume chemistry, and (c) quantitative assessment of enhancements.

3.) The most interesting aspect of this manuscript, from my perspective, is the quantitative assessment of the air quality impacts (i.e., enhancements) over baseline conditions as determined by an air quality model. This section, of all of the sections, provides insights into how well air quality models work and why they need to include biomass burning impacts and long range transport. This said, this aspect of the work is not discussed in great detail, nor does it represent a significant focus of the manuscript. Here are some examples and suggestions.

It might make sense to call out a sub-section on the AURAMS model under section 2.3.

In describing the AURAMS, the authors write, "The reliability of the AURAMS baseline simulation was determined by examining the range of differences between observed and modelled values during non-event days (July 5 and from July 12-16). This range was used to estimate the uncertainty in LRT enhancement (observed – 20 baseline) at each monitoring location." As with any analysis, understanding the null cases (i.e., here how well the model matches observations without smoke plumes) is very important. Where is this analysis described and presented? I do not find data presented on July 5th/12th. Where the uncertainty ranges in relative or absolute units? This portion could use more details.

Has the AURAMS model been used for these types of analysis before? If so, it would be useful to reference previous work.

Question – does the AURAMS model have the capability to include biomass burning plumes? If so, then why was it not? If not, then it would be useful noting this issue, along with the significant differences in the observed and model O3 and PM2.5 in the LFV network at night. While the O3 issue is discussed, the obvious discrepancy in PM2.5 is not discussed. All of these issues add further insight into the issues with current (or at least this particular) air quality model.

The first case presented and discussed, comparing observations to modeled baselines, is the Whistler case shown in Fig. 6. The discussion and quantitative analysis comes from the time period of July 6-8th (WHI1/WHI2). However, it is apparent from

Figure 6 that the PM and O2 are higher than the model baselines from July 6th through July 11th. What is going on during July 8th to 11th? If this is not biomass plume influenced observations, then there is an issue with the concept of model baselines. If it is continued influence of the biomass plume, why is it not apparently included in the analysis and discussions?

The second case, LFV, is shown in Fig.'s 8 and 9. Cottle et al. Fig. 3 shows the lidar smoke plumes in the free troposphere (i.e., above boundary layer) at the CORALNet UBC site on July 6-8th, which becomes the focus of the description in the current manuscript in Fig. 8 with the LFV1/LFV2 boundaries called out. While Cottle notes the entrainment of aerosol into the BL that occurs in the days after this event, the current manuscript does not appear to make this clear direct connection between the lidar observations and the increased PM2.5 measurements (Fig. 8).

Minor comments:

1.) page 1 line 20, "The normalized enhancement ratios..." 2.) page 3 line 7 "... encompassing large parts..." 3.) page 4 line 10+ Much of what is in this paragraph is also in Table 1. Suggest reducing this paragraph and using Table 1 for most of the details. Use this paragraph to discuss specifics, rather than just which instrument measured what. 4.) Figure 6 "(d)" label is missing. 5.) Table 1 is missing filter measurements at Whistler. 6.) Table 3: (a) superscript "a", "b", and "c" in backwards order. What do the "*" mean near central interior station ID's? Right parenthesis in "PM2.5(ug/m3)" is too small. 7.) Table 4: In baseline column, "Historical value (July 6th)" missing "y". 8.) Figure 5: Definitely help if the height (km) were included on y-axis and the specific atmospheric conditions described in the text (page 7: thermal inversion vs stable atm.) were highlighted in figure. Furthermore, UIL and YLW are not labeled on Figure 1, which would also really help. 9.) Section 2.4 is not directly relevant to the manuscript and should be removed or placed in SI.

---

## Author Comment (AC1) · 24 Sep 2016

Author's response to reviewer #2 comments on manuscript acp-2016-302

Reviewer #2 comments

I find this manuscript to be reasonably well written, though the amount of material it is attempting to cover is large, causing a general lack of focus. This manuscript attempts to cover a wide ranging of concepts, including biomass burning plume long range transport observations, boundary layer entrainment processes, air quality conditions over a relatively wide geographical area, air quality exceedances for several Canadian and US standards, biomass burning plume chemical analysis, and an assessment of the enhancements in several air quality measures – using an atmospheric model without biomass burning plume as the baseline. In some senses, this work ties these elements together knowing that this is what it might take to provide fuller understanding and predictive capabilities of the air quality effects of biomass burning events. However, given the amount of material covered and the issues inherent with organization of this material, I feel that it dilutes any overall scientific significance of this particular manuscript. From my perspective, the most informative and useful portion of the manuscript is well summarized in the first two paragraphs of the conclusions. Here, the authors focus on the geographic range impacted by a specific long range transported biomass burning plume, the observed air quality exceedances, observations of the plume chemical characteristics, and comparisons with a baseline model run in an attempt to assess the quantitative impact of the plume on air quality. I expand on these comments below with some suggestions to try and focus the manuscript some and make it more significant.

Author's response:

We thank the reviewer for his constructive suggestions on how to improve the organization and focus of the manuscript. We acknowledge the broad scope has been a challenge and will look to make additional structural improvement in response to your comments to improve the clarity of the manuscript. Responses to your Major comment are listed below.

The topic is timely and appropriate for ACP.
With appropriate attention paid to comments from reviewers, this manuscript should be published.

Major comments:

1.) This work appears to be a direct continuation of the Cottle et al. 2014 Atm Env paper on the same topic, but reprises a lot of what is already in Cottle et al.. For example, there are essentially three Figures that were taken, in part, from Cottle et al. and reproduced in this manuscript. Section 3.1 describes these figures to discuss the transport of the biomass burning across the Pacific. What is not detailed is what is new here on this topic. Perhaps the estimate of >6 days in transport? The authors note that they have done additional back trajectory studies, but it is not clear why. Perhaps due to greater geographical area under study? Do any of these issues make or break the interpretation/conclusions?

The atmospheric transport of the July 6-10 smoke event is covered in Cottle et al using HYSPLIT trajectories, NAAPS ground level smoke concentration forecast. Their HYSPLIT analysis of the event is well done and could be relied upon. The trajectory analysis was completed in support of our analysis of the event and presented in the supplemental material (Figure S1) for completeness. The transport aspect of our submission differ slightly from Cottle in that we include observations of MODIS AOD to describe the plume's transport and describe the associated meteorological scenario. These aspects may not support the interpretation/conclusion directly but do bring additional context to the event and provide perspective on the subsidence/entrainment influences noted in the study. Additional forward trajectories help show that nearby North American wildfire was not affecting our domain (larger than Cottle et al.).

It would make the manuscript more manageable if this portion of the manuscript were removed and the manuscript focused on the geographic range, air quality exceedances, chemical characteristics, and assessment of quantitative impact. For example, section 3.1 could be merged into the introduction in some manner, relying on Cottle et al. to describe the Siberian fires, the transport of plumes across the Pacific, and the general region in Canada/US of impact, in detail. Figure 2 could be removed or placed in SI as it duplicates a portion of a Figure from Cottle and focuses only on transport across the Pacific.

Author's response:

We will reframe the transport and meteorology aspects of the manuscript to leverage more from Cottle et al..

2.) The results and discussion section of the manuscript is organized more or less along geographic lines of specific air quality networks (i.e., Whistler, Lower Fraser Valley, etc.), but each section has different information discussed therein, based on the measurements available at a given site/network. The section 3.2 "July 6-10 smoke event overview" contains most, but not all of the discussion of the geographic extend and boundary layer entrainment. This organization leads to Figure 3 connecting satellite observations to time and geography, Figure 4 showing overview of the enhancements of air quality observations compared with modeled baseline conditions which are discussed in more detail later, Figure 5 showing entrainment conditions, more related to Fig. 3 than Fig. 4, all followed by sections on geographic locations and more details on the air quality modeling results. This puts the detailed discussions of plume chemistry and quantitative assessment of air quality parameter enhancements at the end of the figures and mixed in with geographic details. Furthermore, while the data was collected for specific networks of sites (i.e., Washington State in US, LFV, Whistler, and Interior), the discussion of the results suggest that the real impact was limited on coastal areas (i.e., parts of Washington State and LFV) and more significant on the in-land sites (interior and Whistler). Thus, the discussion/organization could focus more on the geographic effects rather than the specific network site locations.

Another potential organization might setup sections focusing on (a) plume impacts by geography (i.e., time and

space), boundary layer entrainment, and air quality observations, (b) plume chemistry, and (c) quantitative assessment of enhancements.

Author's response:
Agreed. A joint coast to inland discussion would be an improvement. Your second suggestion may also work however illustrating the AQ observation early helps shows the exceptional AQ conditions associated with the event and helps support the discussion around baseline modeling.

3.) The most interesting aspect of this manuscript, from my perspective, is the quantitative assessment of the air quality impacts (i.e., enhancements) over baseline conditions as determined by an air quality model. This section, of all of the sections, provides insights into how well air quality models work and why they need to include biomass burning impacts and long range transport. This said, this aspect of the work is not discussed in great detail, nor does it represent a significant focus of the manuscript.

Author's response:
The air quality model did provide a method to account for the air quality at each monitoring location for changes from anthropogenic sources. An assessment of the model performance for non-event days was examined to establish the uncertainty values for the enhancement estimates. Alternatively, baseline conditions could have been estimated based on recent conditions (persistence), climatology, or another model (empirical) however were likely to introduce greater uncertainty in the estimated contributions of the wildfire smoke contributions given the stagnant meteorological conditions. The focus of the paper was on the particular event and the associated air quality impacts. The need for the inclusion of wildfire sources in the air quality model stems in part from the scale of the impacts. There could be a better link provided on how having wildfire smoke sources within the air quality model could lead to better air quality predictions. The scaled smoke event enhancement shows that a major source of pollutants for these air quality exceedences was missing. However, another justification made would be for improved predictability in the degree on entrainment of smoke plume over stagnant/stable airmass and other meteorological interactions.

Here are some examples and suggestions.
It might make sense to call out a sub-section on the AURAMS model under section 2.3.

Author's response:
Section 2.3 will be added to help describe AURAMS.

In describing the AURAMS, the authors write, "The reliability of the AURAMS baseline simulation was determined by examining the range of differences between observed and modelled values during non-event days (July 5 and from July 12-16). This range was used to estimate the uncertainty in LRT enhancement (observed – baseline) at each

monitoring location." As with any analysis, understanding the null cases (i.e., here how well the model matches observations without smoke plumes) is very important. Where is this analysis described and presented? I do not find data presented on July 5th/12th. Where the uncertainty ranges in relative or absolute units? This portion could use more details.

Author's response:

The methodology was described in a very general way by the statement listed above however we recognized the lack of information on the July 5[th] and July 12-16 conditions limits the transparency of the manuscript. Additional analysis conducted on baseline case will be clarified for the next revision. Some of the non-event period was omitted from the analysis to account for other environmental factors (local smoke in the Interior, fireworks on July 4[th]/5[th] in Washington State) and are also listed in the supplemental information.

The uncertainty ranges in Table 4 are in absolute units (ppbv for ozone and ug/m$^3$ for particulate matter). These were calculated by adjusting the observed enhancements (obs – model) with the maximum and minimum bias observed during baseline period for each station. As part of our analysis on baseline conditions, we also calculated a number of other summary statics on the model bias, although it was decided that the maximum and minimum value would provide the most prudent enhancement estimates. These summary statistics are provided in the attached spreadsheet.

Has the AURAMS model been used for these types of analysis before? If so, it would be useful to reference previous work.

Author's response:

The utility of the AURAMS as a baseline model is supported by its use as an air quality research model ( Stroud et al., 2008, Markovic et al., 2011, Stroud et al., 2015 [see below for full references]).  CMAQ and AURAMS comparisons made in the Pacific Northwest, described by Makar et al, 2014 show comparable performance statistics.  I am unaware of other smoke attribution studies that used chemical transport models (AURAMS or otherwise) as the baseline conditions.  Typically, climatology or persistence assumption is used to develop the expected air quality condition in the absence of the event however uncertainty in the estimated baseline is not always examined in those cases.  Another approach would be to use an empirical modeling approach to estimate conditions.

Question – does the AURAMS model have the capability to include biomass burning plumes? If so, then why was it not? If not, then it would be useful noting this issue, along with the significant differences in the observed and model O3 and PM2.5 in the LFV network at night. While the O3 issue is discussed, the obvious discrepancy in PM2.5 is not discussed. All of these issues add further insight into the issues with current (or at least this particular) air quality model.

Author's response:

At the time of study, AURAMS was available and it does not consider the impacts of biomass burning emissions. AURAMS was able to describe the enhancements to ozone and particulate matter examined in this study within a reasonable margin of uncertainty. The study findings could provide a good base analysis on which to compare against future modeling scenarios of the wildfire event as the reviewer has suggested.

The discrepancies between the PM2.5 observations and the baseline model increase in the July 8-10 period beyond our study's measure of the models variability and in a way that is consistent with the findings of Cottle et al. We can make more explicit mention on the PM2.5 deviation from baseline next revision.

The first case presented and discussed, comparing observations to modeled baselines, is the Whistler case shown in Fig. 6. The discussion and quantitative analysis comes from the time period of July 6-8th (WHI1/WHI2). However, it is apparent from Figure 6 that the PM and O2 are higher than the model baselines from July 6th through July 11th. What is going on during July 8th to 11th? If this is not biomass plume influenced observations, then there is an issue with the concept of model baselines. If it is continued influence of the biomass plume, why is it not apparently included in the analysis and discussions?

Author's response:

The discussion focused on the July 6-8[th] periods as this is where the majority of the Siberian wildfire enhancements occurred. Lesser enhancement may have also occurred on July 8-9. Known biomass burning smoke of local origin influenced the findings for July 10[th] (see Fig. S9). Baseline uncertainty estimates from non-event days excluded periods of known North American smoke influence.

The second case, LFV, is shown in Fig.'s 8 and 9. Cottle et al. Fig. 3 shows the lidar smoke plumes in the free troposphere (i.e., above boundary layer) at the CORALNet UBC site on July 6-8th, which becomes the focus of the description in the current manuscript in Fig. 8 with the LFV1/LFV2 boundaries called out. While Cottle notes the entrainment of aerosol into the BL that occurs in the days after this event, the current manuscript does not appear to make this clear direct connection between the lidar observations and the increased PM2.5 measurements (Fig. 8).

Author's response:

PM2.5 did rise after July 8[th] and we can make specific mention that Cottle indicate entrainment followed. Cottle et al described increase fine mode fraction and additional lidar signatures that support the entrainment of PM2.5 within the LFV. Our results confirm and quantify the timing, spatial distribution, and magnitude of those enhancements. The higher enhancement over the northern slope of the LFV suggests the entrainment process was aided by interactions between the smoke plume and the local mountains.

Minor comments:

1.) page 1 line 20, "The normalized enhancement ratios. . ."

2.) page 3 line 7 ". . . encompassing

large parts. . ."

3.) page 4 line 10+ Much of what is in this paragraph is also in Table 1. Suggest reducing this paragraph and using Table 1 for most of the details. Use this paragraph to discuss specifics, rather than just which instrument measured what.

4.) Figure 6 "(d)" label is missing.

5.) Table 1 is missing filter measurements at Whistler.

6.) Table 3: (a) superscript "a", "b", and "c" in backwards order. What do the "*" mean near central interior station ID's? Right parenthesis in "PM2.5(ug/m3)" is too small.

7.) Table 4: In baseline column, "Historical value (July 6th)" missing "y".

8.) Figure 5: Definitely help if the height (km) were included on y-axis and the specific atmospheric conditions described in the text (page 7: thermal inversion vs stable atm.) were highlighted in figure. Furthermore, UIL and YLW are not labeled on Figure 1, which would also really help.

9.) Section 2.4 is not directly relevant to the manuscript and should be removed or placed in SI.

Author's response:

6) will fix a, b, c subscript order. "*" redundant with subscript "a" and should be removed.

8) Having height label on the Skew-T would be challenging as the display is configured based on a standard atmospheric sounding visualization. A rough rule of thumb is that pressure decrease by 100 hPa for every vertical km in the atmosphere up to 500 hPa. I can add the site label and valid time as a label to each plot if needed. Currently labeling seems reasonably clear.

---

## Referee Comment (RC2) · Anonymous Referee #4 · 16 Nov 2016

Reviewing Teakles et al. "Impacts of the July 2012 Siberian Fire Plume on Air Quality in the Pacific Northwest"

The manuscript by Teakles et al. looked at the impact of the July 2012 Siberian wildfires on air quality in the Pacific Northwest regions using various resources such as surface in-situ and satellite remote observing observations, HYSPLIT air trajectory model and a chemical transport model. I find this study is valuable as it tries to provide a full scope of the impact of long-range transport of wildfire plume with using the combination of existing observation dataset and numerical models. It covered from its trajectory and chemical analysis to assessment of impact on air quality standards over Pacific Northwest regions. The manuscript is well within the scope of ACP. However, the manuscript requires some revisions before publishing. I have listed my major and minor comments below. When these comments are addressed in the manuscript, I recommend this to be published in ACP.

Major comments:
1. **How well does the AURAMS model capture the observed air quality?** This study used the AURAMS model simulation without Siberian wildfire influence as a baseline, which is a reasonable approach. My main concern is how well the model simulates the observed air quality. Any model deficiency would influence the main results in this study. Please provide statements about the model performance regarding O3, speciated PM and total PM2 in Section 2.2. In addition, I strongly encourage providing an evaluation of model for the period where the Siberian wildfire doesn't impact the Pacific Northwest regions (maybe July 01 to 05?).
2. The manuscript was overall well written. However, some figures and legend should be improved. Particularly, it was hard to follow the discussion on some individual observation sites and geographical impact analysis, as I am not familiar with most of the site/region name. Also, I found several acronyms are used without definition or defined but not used later (e.g., BC in the abstract, AGL, ASL, LFV, CWS). Please improve throughout the manuscript. Please try to improve them throughout the manuscript.

Minor comments:
Table 1:
1) For individual site, please provide lat/lon/alt information.
2) Please either provide a full name of network or change the legend to tell where in texts to find them.
3) it would be helpful if the order of network/site in Table 1 follows the text in Section 2. It would be even more helpful if it were ordered by country and region.
4) For Mr. Rainier, is it Teledyne-API 400 or Teledyne-API T400?
5) In the last row, the second column about SMPS size range is incorrect. It should be 14 m to 572 nm (to be consistent with the texts) or 0.14 um to 0.57 um.

Table 2 :
1) The legend says " used in the study", but the main text (P5; L13-14) says CWS for O3 were not used in this study. Please fix the inconsistency.
2) Please put reference in new column and get rid of footnote.

Table 3 – Please provide lat/lon/alt for each sties.
Table 4 –The current version is not easy to read due to multiple lines in "area of interest".  I understand the table may look different for the published version. Please check the table readability again for published version.
Table S1- please try to use the same full/acronym name for network/sites as Table 1.

Figure 1
1) Please use numbering for each site in a map and provide a list of full and short name for each site. The main texts often use full site names but Fig 1 shows short name only.
2) Provide wildfire location in Fig 1a.
3) Please improve figure legend; including the black outlined area (LFV); changing "Lower Fraser Valley" to "Lower Fraser Valley (LFV)"; explaining the symbol type.
4) Please include all the sites that are used in the texts in Figure 1. The main texts uses more sites than presented in Fig 1. I was quite confused which site they are talking about. Also, please provide lat/lon/height for each site when it is first mentioned in the texts.
5) Regarding the Section 3.3 to 3.6, it would be helpful to see where they are located in a map.

Figure 2
1) The MODIS true color image is hard to see. Not sure where (a) is over exactly.
2) Please consider moving the second sentence to acknowledgements.
3) I'd like to see the AOD plots like (b)-(d) but from July 1$^{st}$ to 6$^{th}$ as a consecutive order in supplementary materials.  That would be more convincing to show the plume transport.

Figure 3
1) This figure needs much improvement. It is really hard to see the smoke plume.
2) What is the light blue circle near the KFS site?
3) For 24-hr PM2.5, is it just daily mean? Similarly for Fig 4, what do you mean by "maximal" enhancement of 24-hr PM2.5?

Figure 4
1) Please provide the temporal period used in the figure.
2) Similarly to Fig 1, I strongly recommend to put a number in each site.

Figure 5
3) I don't see the two sites in Fig 1. Please show them in Figure 1. Also, please put lat/lon/alt information for each site here. The texts explaining Fig 5 uses height but the figure shows pressure. Please provide a pressure level for the height discussed in the main texts.

4) In the legend, "dry bulb" to "dry bulb temperature".

Figure 6
1) Please explain WHI1 and WHI2 in the legend.
2) In the legend (3rd line), "a horizontal blue line" should be "a horizontal white line". I see white line.
3) "(d)" must be shown in the PM figure. "ACSM" line is not explained in the legend.

Figure 7
1) Why is the hourly organics lower than OC(TOT) some period?
2) Is there any particular reason to use "particle SO4" instead of just "so4"? Here is all about aerosol chemistry, so it sounds a bit odd to call "particle SO4".

Figure 8
3) Related to my first major comment, the model doesn't seem to capture the observed PM2.5 before July 8th. Please see if the model has systematic biases in PM simulations. Given that, please provide how it may affect the results.

Figure 9 – Unless what the text explains about Fig 9c (P9;L28-31), I don't see any clear increase in O3 episode in Figure 9c.

Figure 10 – Please provide lat/lon/alt for each site.

Figure S1- Please mark the wildfire locations.

Figure S2 – What period is it? And please put time and location information for the wildfires.

Figure S3- Please use full name of MSLP.

Figure S5 – I don't understand why this figure reflects the OC dominance.

Figure S6 – I am not sure the plot and legend are consistency. Please check the legend again and consider rewrite it.

Main texts

P2 L29-30 – Please provide a reference

P2 L30 – Please add year (2012) after "July and August"

P2 L31 - Please add year (2012) after "August

P3 L5 – Please explain more what you mean by "noted entrainment signature".

Section 2.1.1 – Please keep the same order as Table 1 and provide lat/lon/alt information.

Section 2.1.2 – Table 1 shows CORALNet, which is not mentioned in the text here.

Section 2.1.3 – Isn't this part of Ambient AQ monitoring data? If so, perhaps move to Section 2.1.2.

P5 L6 – Is there any good reason for choosing 10m, 2.5km, 5km, and 7.5km? It seems too big jump from 10 m to 2.5km.

P5 L9 – Are all 72 sites shown in Figure 1?

P5 L27-28 – What do you mean by interpolate weather into the AURAMS domains? Not dynamically computed? Does it mean it is subject to a potentially large bias?

P6 L1 – What is the size range covered in AURAMS?

P6 L3-8 – This is related to my first major comment. Can you comment on any expected bias when using climatology as boundary condition? Again, the model evaluation should be presented in this study.

P6 L23-25 – how far were the Siberian wildfire plumes rise?

P7 L3 –please provide the NA wildfire period.

P7 L4-7 – It is hard to understand what Figure S3 shows.

P7 L11 – I am not sure if I missed something, but I don't see PM increase for Whisler High Elevation site.  Please put numbers for each site in Figure 4. It should help to understand the texts better.

P7 L15 and L17 – Please provide pressure level for that height.

P7 L10-11 – I don't understand why Figure S5 reflects that OC is dominant during that period.  Any explanation?

P8 L11 – What is "1 hour data"? hourly mean?

P9 L18 – "High O3 concentration" ➔ "High O3 concentrations"

P9 L18-19 – Is this also due to the Siberian fires?

P11 L3-7 – These parts leaved me more questions than answer. Can you explain more about the surface PM2.5 analysis and impact analysis?

P11 L15-16 –If I understood this correctly, " the additional" is better to be removed.

P11 L23-25 – To be consistent, please don't use bracket.

P11 L32 – I believe most CTM simulations include wildfire emissions. The recommendation doesn't sounds useful. Please clarify it if necessary.

---

## Author Response (AR1)

**Response to reviewer #4**

**Major comments:**

**1. How well does the AURAMS model capture the observed air quality?**
This study used the AURAMS model simulation without Siberian wildfire influence as a baseline, which is a reasonable approach. My main concern is how well the model simulates the observed air quality. Any model deficiency would influence the main results in this study. Please provide statements about the model performance regarding O3, speciated PM and total PM2 in Section 2.2. In addition, I strongly encourage providing an evaluation of model for the period where the Siberian wildfire doesn't impact the Pacific Northwest regions (maybe July 01 to 05?).

Author's response:

The quality of the baseline model used in the study is a fair concern. The accuracy of the baseline simulation is a limiting factor in resolving air quality enhancements especially in regions like the LFV where the enhancements were shown to be less pronounced. Our study carefully assessed the reliability of the model simulation on a site-by-site basis and included an appropriate uncertainty measure to the 8-hr $O_3$ and 24-hr $PM_{2.5}$ enhancement reported in Table 4. A bias in the baseline simulation could affect the reported enhancement result; however, the actual enhancement (not known) should fall into the range of enhancement listed in Table 4. Other baselines derived from climatology or persistence forecasts are also subject to the same uncertainties and constraints but are less often explored in that context. Generally, air quality predictions from air quality models outperform simple climatological and persistence forecasts particularly under changing underlying air quality conditions (air quality episodes).

Additional information regarding the overall performance of the AURAMS baseline simulation during non-event days (July 5[th] and July 12[th] to 15[th]) was added to Section 2.3.1 AURAMS baseline simulation and to the supplementary material as Table S2. The 1-hr model performance for O3 and PM2.5 is comparable to the assessed reliability of the AURAMS model from other studies (Makar et al., 2014) (Table S2). Please note the model performance for 8-hr and 24-hr averaging periods could differ from the 1-hr statistics but are reflected in the uncertainty ranges in Table 4. The 8-hr and 24-hr uncertainty ranges have also been added to Table S1 for selected sites. The baseline simulation of speciated PM2.5 would have no effect on the findings of the study as these fields were not examined.

2. The manuscript was overall well written. However, some figures and legend should be improved. Particularly, it was hard to follow the discussion on some individual observation sites and geographical impact analysis, as I am not familiar with most of the site/region name. Also, I found several acronyms are used without definition or defined but not used later (e.g., BC in the abstract, AGL, ASL, LFV, CWS). Please improve throughout the manuscript. Please try to improve them throughout the manuscript.

Author's response:

Improvements to the geographical referencing of sites and regions have been improved thanks to the corrections for the minor comments listed below.   Figure 1 has been modified to use numeric site ids for ambient monitoring sites listed in Table 3 and lettered site id for other sites mentioned in the study. Supplementary information on the station names, ids, latitude, longitude, and altitude has been added to Table S1. Site ids have also been added to Figure 2 and 3.  MODIS image showing wildfire location in Russia and smoke drifting eastward across the Pacific has now been incorporated into Figure 1 and georeferenced using a bounding box on Figure 1a.  The polygons for the province of British Columbia, Canada Washington State, United States of America have been highlighted on Figure 1 for added clarity on the regions affected by the smoke plume.  The location of the Kelowna and Quillayute airports were also noted in Figure 1.  Please see response to minor comments below for added details.

Acronyms for AGL and ASL are now defined in the text and captions when used.

**Minor comments:**
Table 1:

1.  For individual sites, please provide lat/lon/alt information

    Author's response: Lat/lon/alt information was added as a new table (Table S1) in the supplementary section.

2.  Please either provide a full name of network or change the legend to tell where in the texts to find them
    Author's response: Revised Table 1 to include the full name of the network

3.  It would be helpful if the order of network/site in Table 1 follows the text in Section 2. It would be even more helpful if it were ordered by country and region.
    Author's response: Revised Table 1 so that Networks/Site are ordered by Region and then by order in the text.

4.  For Mr. Rainier, is it Teledyne-API 400 or Teledyne-API T400?
    Author's response: Corrected in the latest version of Table 1 to Teledyne-API T400.

5.  In the last row, the second column about SMPS size range is incorrect. It should be 14 m to 572 nm (to be consistent with the texts) or 0.14 um to 0.57 um.

Author's response: Text revised to indicate a size range of 14 nm to 572 nm. Equivalently, it would be 0.014 um to 0.572 um.

Table 2: The legend says "used in the study", but the main text (P5; L13-14) says CWS for O3 were not used in this study. Please fix the inconsistency.

Author's response:

Removed the text "used in the study" from the legend.

Main text (P5; L13-14) was meant to indicate that multi-year aspect of CWS was not consider even though the 8-hr threshold used by CWS was a benchmark for severely degraded air quality conditions in the study.

Main text (P5; L12-17) has been revised to the following to improve clarity:

"Based on measurements collected over 70 air quality monitoring stations in the Pacific Northwest, the air quality impacts of the Siberian wildfire plume were assessed by examining the $O_3$ and $PM_{2.5}$ concentrations at various averaging periods of 1-hr, 8-hr and 24-hr, depending on the pollutant. For each monitoring station, severe air quality episodes were identified using the regional objective and national standards, applicable at the time of the event (Table 2), as thresholds, and were compared to the average historical July daily maxima values from 2000 to 2010. It should be noted that although some of these standards, for example the Canada Wide Standard (CWS) for O3 (CCME, 2014), are based on multi-year statistics, all standards were compared to hourly observations of their respective averaging periods."

Please put reference in new column and get rid of footnote.

Author's response: New reference column added to Table 2.

Table 3 – Please provide lat/lon/alt for each site.

Author's response: Lat/lon/alt was not directly listed for all the 50+ sites used in the study. This information is available through the source information provided in Table S3 (previously Table S1) and also provided in a new table (Table S1) in the supplementary section. Table 3 was revised to incorporate the site id used in Figure 1.

Table 4 –The current version is not easy to read due to multiple lines in "area of interest". I understand the table may look different for the published version. Please check the table readability again for published version.

Author's response: Modified Table 4 to have a wider" area of interest" column

Table S1- please try to use the same full/acronym name for network/sites as Table 1.

Author's response: Table S1 has been adjusted to use the same network names as Table 1 and to make a few minor formatting changes

Figure 1
1) Please use numbering for each site in a map and provide a list of full and short name for each site. The main texts often use full site names but Fig 1 shows short name only.
Author's response:  A numbering approach was adopted for labelling sites on Figure 1 mentioned in Table 3. Other sites including lidar, atmospheric radiosondes, marine and background monitoring are lettered from A to G.  The full site name and lat/lon/alt information are provided as a new table (Table S1) in the supplementary section.    Short names for sites were removed as they were rarely used and no longer needed.

2) Provide wildfire location in Fig 1a.
Author's response: Siberian wildfires were widespread across Eastern Russia at the time of the event and smoke originated from an aggregate of these active fires.  MODIS true color image (previously part of Fig. 2) is now incorporated into Figure 1(b). A boundary box is added to transport map to show the location of the MODIS image.

3) Please improve figure legend; including the black outlined area (LFV); changing "Lower Fraser Valley" to "Lower Fraser Valley (LFV)"; explaining the symbol type.
Author's response: The Lower Fraser Valley was changed to Lower Fraser Valley (LFV; outlined in black in Figure 1(c)).

4) Please include all the sites that are used in the texts in Figure 1. The main texts uses more sites than presented in Fig 1. I was quite confused which site they are talking about. Also, please provide lat/lon/height for each site when it is first mentioned in the texts.
Author's response: All the sites that are explicitly mentioned in the text have now been added to Figure 1. The lat/long information for these sites is now presented in supplemental Table S1.

5) Regarding the Section 3.3 to 3.6, it would be helpful to see where they are located in a map.
Author's response: Currently this is not listed in the manuscript; however, Table 3 does provide an overview of the stations contained in these regions and as such provide sufficient geographical information for the purpose of this study.

Figure 2
1)   The MODIS true color image is hard to see. Not sure where (a) is over exactly.
Author's response: MODIS true color image is now incorporated into Figure 1(b). A boundary box is added to Figure 1(a) to show the location of the MODIS image.

2)   Please consider moving the second sentence to acknowledgements.
Author's response: This is now reflected in the new Figure 1 caption.

3)   I'd like to see the AOD plots like (b)-(d) but from July 1st to 6th as a consecutive order in supplementary materials. That would be more convincing to show the plume transport.
Author's response: Figure 2b to d and additional AOD plots are now incorporated in the supplementary material as Fig S2.

Figure 3
1) This figure needs much improvement. It is really hard to see the smoke plume.
Author's response: The main focus of figure 3 is to illustrate the worst air quality conditions experienced by the network on a daily basis. The MODIS image was added to supplement the discussion of the air quality conditions experienced by the network.  MODIS AOD images for July 6th, 8th, and 10th in Fig S2 will be used to support the discussion of plume transport in the Pacific Northwest.

2) What is the light blue circle near the KFS site?
Author's response: The light blue circle near the KFS site is part of the MODIS image product used.  It marks the Kelowna location geographically and is part of the tailored MODIS imagery available for this location.  Figure 2 caption has been revised to include:
"The light blue circle on the MODIS source imagery indicates the location of the Kelowna Airport (site G) on all panels. "

3) For 24-hr PM2.5, is it just daily mean? Similarly for Fig 4, what do you mean by "maximal" enhancement of 24-hr PM2.5?
Author's response: For 24-hr PM2.5, it is not just the daily mean.  Instead, we calculate the observed PM2.5 as a rolling 24-hr average metric and report the peak values on a given day.  This metric provide a more reliable view of the peak air quality conditions from the event since the smoke plume had only a temporary effect on some sites.  This avoids time issues when the plume enhancement lasted < 24 hours and stretched between 2 calendar days.

For Figure 4 (now Figure 3), we compare the rolling 24-hr PM2.5 condition to the baseline condition. The maximal difference between the two indicates how much the additional PM2.5 from the wildfire smoke affected a sites air quality.

Figure 4
1) Please provide the temporal period used in the figure.
Author's response: We modified the figure caption to read:

*"Fig.  3.  Overview of the maximal enhancements to 8-hr O3  and 24-hr PM2.5 rolling average estimated for the study area (a and b, respectively) and the LFV (c and d, respectively) based on the differences between the ambient air quality observation and the AURAMS baseline modelling for the July 6-10, 2012 smoke event. Abbreviations used in this figure are defined in Table S1 and throughout the text."*

2) Similarly to Fig 1, I strongly recommend to put a number in each site.
Author's response: We've added numbers and/or alphabetical letters to stations that are explicitly mentioned in the text.

Figure 5
3) I don't see the two sites in Fig 1. Please show them in Figure 1. Also, please put lat/lon/alt information for each site here. The texts explaining Fig 5 uses height but the figure shows pressure. Please provide a pressure level for the height discussed in the main texts.
Author's response: Kelowna Airport, BC (Site G) and Quillayute Airport, WA (Site E) have been added to Figure 1 and identified as radiosonde sites in the legend.  The lat/lon/alt metadata for these stations has been included in a new Supplemental Table S1.

The titles for the subpanels in Figure 5 (now Figure 4) have been adjusted to match the station name found in Table S1.

Mixing height is typically reported in height coordinates whereas tephigrams are rely mainly pressure coordinates. Height estimates can be used instead if a standard atmosphere is assumed. To address the discrepancy, the level at which the mixing height was reached is now indicated on each subpanel with a label indicating the height in meters.

The caption was revised to:
"Fig. 4. Skew-T atmospheric radiosondes for Quillayute (UIL) and Kelowna (YLW) at 00Z on July 8th, 2012. The atmospheric dry bulb temperature (T) and dewpoint temperature (Td) are shown from the surface up to 100 mb. Black dashed line is a trace of a surface parcel lift to estimate the mixed layer depth at the height where it intersects the environment dry bulb temperature."

4) In the legend, "dry bulb" to "dry bulb temperature".
Author's response: Caption now includes dry bulb temperature (see above).

Figure 6
1)   Please explain WHI1 and WHI2 in the legend.
Author's response:
WHI1 and WHI2 are described in the revised caption.
"Fig. 5. Whistler lidar (site A, 650 m ASL) measured backscatter ratio of the 1064 nm channel to that of clear air for the July 2012 Siberian smoke plume (a) and observed (solid) 1-hr O3 (b), 1-hr CO (c), and 1-hr PM1 (based on the ACSM) and PM2.5. (d) at Whistler Peak High Elevation station (site 1, 2182 m ASL, indicated as a horizontal white line in panel a). AURAMS baseline (dashed) 1-hr O3 and 1-hr PM2.5 are shown in panel b) and d), respectively. Elevated aerosol backscatter in panel (a) are noted from July 6 14:00 PST (WHI1) to July 8 06:00 PST (WHI2) by red dashed lines. The shaded regions indicate night time hours between 18:00 PST to 07:00 PST."

2)   In the legend (3rd line), "a horizontal blue line" should be "a horizontal white line". I see white line.
Author's response: Revised caption to read white line (see above).

3) "(d)" must be shown in the PM figure. "ACSM" line is not explained in the legend.
Author's response: ACSM now mentioned for PM1 mass estimates (see above).

Figure 7
1) Why is the hourly organics lower than OC(TOT) some period?
Author's response: There are a few reasons why the hourly organics can differ from OC(T900; previously TOT). The first is that the OC(T900) and hourly organics from the ACSM have different sampling periods. The OC (T900) is derived from two 8-day integrated samples (day and night). Secondly, it is estimated that a multiplier (or ~1.9) is needed to convert OC to OM for an aged aerosols. Other considerations are some ACSM calibration issues and loss of semi-volatiles due to the fact that EnCan-total-900 is a thermal method that purges samples to 50 C.

To clarify the sampling period of the EC(T900) and OC(T900), we have adjusted the caption to re-iterate the sampling period for the instrument even, although it was mentioned in the methodology section.

Revised caption reads:

"Fig. 6. Aerosol chemistry measurements taken at Whistler Peak High Elevation site (site 1) between July $6^{th}$ 2012 and July $12^{th}$, 2012. Hourly Organics (Org), $NO_3^-$, $NH_4^+$, $SO_4^{2-}$ measured by the ACSM (a, b), EC and OC as sampled over a 8-day period (July 3-11, with day and night split) using the EnCan Total-900 thermal method (T900) (a), SO4 as a 3-day integrated sample by quartz filter pack (FP) (b), and rBC as acquired using the Single Particle Soot Photometer (SP2) (c). The shaded regions indicate night time hours between 18:00 PST to 07:00 PST."

2) Is there any particular reason to use "particle SO4" instead of just "so4"? Here is all about aerosol chemistry, so it sounds a bit odd to call "particle SO4".
Author's response: Revised legend and caption to use $SO_4^{2-}$.

Figure 8

Related to my first major comment, the model doesn't seem to capture the observed PM2.5 before July 8th. Please see if the model has systematic biases in PM simulations. Given that, please provide how it may affect the results.
Author's response: The AURAMS baseline simulation is having some challenges reproducing the hourly $PM_{2.5}$ trends in the LFV. Figure 8 b shows the LFV regional bias for 1-hr $PM_{2.5}$ centered on the zero mark with network average deviations between +3 and -5. However, the modelled 24-hr $PM_{2.5}$ metric is somewhat more reliable as it smooths out the peak in through in the model performance. The enhancement to the 24-hr PM2.5 for LFV network, listed in Table 4, reflects an enhancement ranging from 6 to 13 ug/m$^3$ (or an uncertainty of -3 to +4 in the 24-hr $PM_{2.5}$). The minimum enhancement of 6 ug/m$^3$ indicates that model uncertainty cannot account for the variance 24-hr $PM_{2.5}$ conditions that occurred during the smoke event. Additional discussion on 1-hr PM2.5 model performance is discussed in the new subsection 2.3.1.

Figure 9 – Unless what the text explains about Fig 9c (P9;L28-31), I don't see any clear increase in O3 episode in Figure 9c.
Author's response: Figure 9c (now Figure 8c) shows that the O3 exceedance that occurred on July 8 2012 in Chilliwack may have been solely due to anthropogenic sources and is not meant to be reflective of the overall enhancement observed in the LFV, which was illustrated by Figure 9a (now Figure 8a). In addition, on July $9^{th}$ and $10^{th}$, enhancement to the 8-hr daytime O3 varies spatially across the LFV as describe by Figure S6, with the strongest enhancement over the northern portion of the LFV and closer to the coast. Table 4 indicates a daily maximum enhancement to the 8-hr O3 in the LFV of 10 ppbv with an uncertainty range of 2 to 23 ppbv (-8 to +13).

Figure 10 – Please provide lat/lon/alt for each site.
Author's response: These are now incorporated in supplemental Table S1.

Figure S1- Please mark the wildfire locations.
Author's response: Figure S1 removed has been removed and greater emphasis on the transport assessment of the Siberian plume has been made in Introduction.
Modis True Colour Images of wildfires in Siberia on June $29^{th}$ have been added to Figure 1 and georeferenced using a bounding box on panel (a) of Figure 1.

Figure S2 – What period is it? And please put time and location information for the wildfires.

Author's response: The release date/time for the forward trajectory modeling for the Long Draw and Waldo fires has been added to the Figure S2 (similar to the previous S1) and caption has been revised to lists the release date/time with the release height information.

Caption revised to:
"Fig. S1. Five day forward trajectories using the CMC Trajectory model from the Long Draw, Oregon (a) and Waldo Canyon, Colorado wildfires (b) released at 00UTC on July 9$^{th}$, 2012 at heights of 10 m, 100 m, and 1 km AGL."

Main text revision in section 2.2 also reflects the release time for the trajectory modeling.

1) Waldo Canyon, Colorado wildfire: detected June 23, 2012 and was declared 100 percent contained on July 10, 2012
2) Long Draw, Oregon wildfire: started July 8$^{th}$ at 6pm by lightning and grew quickly.

Figure S3- Please use full name of MSLP.
Author's response: Caption has been revised to reflect the added AOD source imagery added:
"Fig. S2. The daily average MODIS Aerosol Optical Depth product for July 1-6 (a to f) illustrates the eastward progression of the Siberian Fire plume in early July onto the Western Pacific prior to its arrival off the coast of Washington and Oregon States on July 6$^{th}$, 2012 (f). Enhanced AOD values spread across South and Central British Columbia by July 8$^{th}$, 2012 (g) then shift southeastward out of the Pacific Northwest domain on July 10$^{th}$, 2012 (h). Mean sea level pressure is contour (in solid black) on all panels.."

Figure S5 – I don't understand why this figure reflects the OC dominance.
Author's response: Sentence is revised to "Further, a comparison of the mass concentrations estimated from the SMPS, based on the assumptions of spherical particles and a particle mass density of 1.2 g/cm$^3$ as a result of the dominant organic composition during that period (Figure 6), with the ACSM for July 9-31 is illustrated in Figure S4."

Figure S6 – I am not sure the plot and legend are consistency. Please check the legend again and consider rewrite it.
Author's response: Unsure on how to revise the legend at this time but remain open to further revision prior to final publication

**Main texts**

P2 L29-30 – Please provide a reference
Author's response: No reference is needed in this case. The sentence has been revised for clarity to: Further study is still required to better understand the interactions of the above factors on downwind air quality for long-range transport events".

P2 L30 – Please add year (2012) after "July and August"
Author's response: Modified to "July and August 2012".

P2 L31 - Please add year (2012) after "August

Author's response: Modified to "August 2012".

P3 L5 – Please explain more what you mean by "noted entrainment signature".

A "noted entrainment signature" is meant to indicate a period where aerosols are entrained into the boundary layer through turbulent or thermal mixing processes.  Cottle et al noted the entrainment of layers aloft using increased optical thickness measurements in the boundary layer over time, low volume depolarization levels aloft and in the boundary layer.

Author's response:

The main text on P3 L4-6 was revised to clarify to:

"HYSPLIT trajectory analysis confirmed that aerosol layers subsiding over the LFV had largely originated from over wildfire source areas in Siberia at least 6 days earlier.  Aerosol backscatter measurement and low depolarization volume ratios during the event showed the progressive entrainment of smoke into the LFV through July 10th, 2012. Their findings suggest that the high PM2.5 observed by the region's fixed air quality monitoring network was associated the progressive entrainment of aerosols into the LFV."

Section 2.1.1 – Please keep the same order as Table 1 and provide lat/lon/alt information.

Author's response: Table 1 has been revised to order the sites by geographical region and then by order presented in the text.

Section 2.1.2 – Table 1 shows CORALNet, which is not mentioned in the text here.

Author's response: CORALNet was the previous terminology used when referencing the lidar network but is no longer used or referenced in that way.  Originally, the main text and Table 1 incorporated this information but was then revised in later version of the manuscript.

Table 1 has been revised to no longer reference CORALNet.

Section 2.1.3 – Isn't this part of Ambient AQ monitoring data? If so, perhaps move to Section 2.1.2.

Author's response: The instrumentation and purpose for the Whistler High Elevation Monitoring Site differs from that of the Ambient AQ monitoring data listed in section 2.1.1 and 2.1.2.   These differences are also consistent with categories and labeling in Figure 1.

P5 L6 – Is there any good reason for choosing 10m, 2.5km, 5km, and 7.5km? It seems too big jump from 10 m to 2.5km.

Author's response: Figure S1 has been removed.  Introduction and Section 2.2 has been modified to rely on Cottle et al. transport assessment for the plume.

The previous logic for the release heights is somewhat arbitrary and essential a desired to estimate the transport for various heights in the atmosphere ranging from near the surface to near the tropopause.  For our study 4 evenly spaced heights were used.  Wildfire aerosols lofted above 5km are transported to the Pacific Northwest within a 7 to 10 day timeframe (based on previous dispersion results).

P5 L9 – Are all 72 sites shown in Figure 1?

Author's response: A total of 73 sites (72 + Whistler High Elevation Monitoring Site (site #1)) were used to determine the enhancements.  Several communities (William Lake (site #9), Quesnel (site #10), Prince

George (site #11) have multiple sites in the community. Due to the scale of the figures 1, 3, 4, only 63 sites plotted in Figure 3 (similar issues exist with over plotting in Figure 1 and 2.

P5 L27-28 – What do you mean by interpolate weather into the AURAMS domains?
Not dynamically computed? Does it mean it is subject to a potentially large bias?
Author's response: Driving meteorological fields were interpolated from the 2.5 km meteorology simulated by the GEM-LAM model to the 4 km grid used in this study. The interpolation is not ideal but commonly done when setting up atmospheric modeling studies on local area model domains. The advantage in this case is the driving meteorology is based on a higher resolution model which can help minimize any information loss. The interpolation was required since both the resolution and domain of the AURAMS simulation differs from the meteorological source data. It does not mean that it is subject to potentially large bias and can be assess on its reliability based on both the model performance statistics provided in Table S1 and Table S2.

P6 L1 – What is the size range covered in AURAMS?
Author's response: AURAMS uses a chemically-speciated 12-bin sectional distribution to characterize particulate matter from 0.01 to 40.96 µm in diameter. Main text on P6 L1 has been modified to reflect this.

P6 L3-8 – This is related to my first major comment. Can you comment on any expected bias when using climatology as boundary condition? Again, the model evaluation should be presented in this study.
Author's response: A climatological boundary conditions is currently the standard setup for the AURAMS model to use in regional studies. It could lead to a poor performance of the model particularly when conditions at the lateral boundaries deviate from the climatological expectation. This was clearly the case during the July 2012 Siberian wildfire event and justifies statements made in the conclusion section about the need to include these sources to improve air quality modeling. Performance of the AURAMS modeling in this study has now been included through Table S1 and S2, to supplement the existing uncertainty ranges provided in Table 4.

P6 L23-25 – how far were the Siberian wildfire plumes rise?
Author's response: Dispersion modelling suggests that pollutants lofted above 5km reached the Pacific Northwest with 6-10 days.

P7 L3 –please provide the NA wildfire period.

Author's response:
1) Waldo Canyon, Colorado wildfire: detected June 23, 2012 and was declared 100 percent contained on July 10, 2012
2) Long Draw, Oregon wildfire: started July 8[th] at 6pm by lightning and grew quickly.
3) Matthew Creek (49 45.550, - 120 48.396) on July 9[th] 2012

P7 L4-7 – It is hard to understand what Figure S3 shows.

Author's response: AOD imagery for revised Fig. S2 will help with discussion of plume position in section 3.1.  L4-9 revised to

"July 8-9, poor air quality conditions continued to spread across the Interior, coinciding with the northward and inland progression of the plume as collaborated by AOD imagery (Fig. S2 g), and caused air quality exceedances at several communities (see Table 3).  Over in the coastal region, elevated $O_3$ and $PM_{2.5}$ also occurred on July 8[th], with isolated 8-hr $O_3$ exceedances at Chilliwack (site 3) and Enumclaw (site 5).  MODIS AOD on July 10[th] (Fig. S2 h) show the plume remnant had shifted southeastward and out of the Pacific Northwest domain."

In addition, discussion of plume motion relative to meteorological features (L1, L2) on Figure S3 (now Figure S2) were omitted in the latest revision.

P7 L11 – I am not sure if I missed something, but I don't see PM increase for Whistler High Elevation site. Please put numbers for each site in Figure 4. It should help to understand the texts better.

Author's response: 24-hr PM increase at Whistler High Elevation site was 10µg/m$^3$ and is reflected properly on Figure 3. Figure 3 now has labels for sites listed in Table 3 and Table S1.

P7 L15 and L17 – Please provide pressure level for that height.

Author's response: See previous response.  Corrections to Figure 4 should help establish what pressure level we are referring.

P7 L10-11 – I don't understand why Figure S5 reflects that OC is dominant during that period. Any explanation?

Author's response: Sentence is revised to "Further, a comparison of the mass concentrations estimated from the SMPS, based on the assumptions of spherical particles and a particle mass density of 1.2 g/cm3 as a result of the dominant organic composition during that period (Figure 6), with the ACSM for July 9-31 is illustrated in Figure S4."

P8 L11 – What is "1 hour data"? hourly mean?

Author's response: Yes, 1-hr averaged data (ACSM & SMPS are sampled on a half hour interval).

P9 L18 – "High O3 concentration" è "High O3 concentrations"

Author's response: The text was revised.

P9 L18-19 – Is this also due to the Siberian fires?

Author's response: The high ozone levels are likely due to the Siberian Fire.

P11 L3-7 – These parts leaved me more questions than answer. Can you explain

more about the surface PM2.5 analysis and impact analysis?
Author's response: The Firework output was removed in the latest revision.
main text for L3-7, P11 has been revised to

"On July 10th, as the bulk of the Siberian plume shifted northeastward out of British Columbia, remnants of the plume shifted over the Southern Interior and Whistler; however, attribution of the plume's impacts was impeded due to local wildfire activity (Matthew Creek; approx. 100 km west of Kelowna; 155 acres).  Overall, the maximal air quality enhancements due to smoke from the Siberian wildfires in the Southern Interior was approximately 15 ppbv and 15 μg/m$^3$ for 8-hr $O_3$ and 24-hr $PM_{2.5}$, respectively, based on the impact analysis conducted at Vernon (site 7) for the July 6-9 period (Fig. S10)."

P11 L15-16 –If I understood this correctly, " the additional" is better to be removed.
Author's response: "Additional" was removed.

P11 L23-25 – To be consistent, please don't use bracket.
Author's response: The square brackets were removed.

P11 L32 – I believe most CTM simulations include wildfire emissions. The recommendation doesn't sounds useful. Please clarify it if necessary.
Author's response: A climatological boundary conditions is currently the standard setup for the AURAMS model to use in regional studies.  It could lead to a poor performance of the model particularly when conditions at the lateral boundaries deviate from the climatological expectation.  This was clearly the case during the July 2012 Siberian wildfire event and justifies statements made in the conclusion section about the need to include these sources to improve air quality modeling.

Text in conclusion on L31-32, P11 and L1, P12 was revivsed to clarify:

"This demonstrates the need to include wildfire emissions within chemical transport model, and to derive effective parameterizations for the lofting and subsequent reactions in modelling ambient air quality conditions. Lateral boundary for chemistry used in regional air quality modeling application should be responsive to the long-range transport of pollutants."

---

## Author Response (AR2)

**Response to referee #2**

**Minor comments:**

This is the second review on this manuscript. The authors adequately responded to my comments and concerns on the first review by restructuring the manuscript and I find this manuscript almost ready for publication. This manuscript should be published, as it is relevant and well-constructed.

I have a few, very minor comments:

1.) lines 29-30 in the abstract have Whistler Peak listed in two sentences, one describing significant effects and one describing minor effects. It cannot be in both.

> Author's response: Whister Peak listed in error for describing minor effects.
> Abstract text on ln 29-30, pg1 revised to: Lesser enhancements of 10-12 ppbv for 8-hr $O_3$ and of 4-9 µg/m$^3$ for 24-hr $PM_{2.5}$ occurred across coastal British Columbia and Washington State.

2.) The last paragraph (1 sentence) of the introduction is essentially a rewording of the second to last sentence. Suggest removing the last sentence or reducing the wording in the second to last sentence.

> Author's response: Amalgamated text and themes from the last two sentences together.
> Introduction text on ln 10-14, pg 3 revised to
> " *This study expands on the Cottle et al. (2014) work in a number of significant ways: it analyzes potential air quality impacts over a much greater geographical area encompassing large parts of British Columbia and Washington State; it uses detailed air quality measurements at a high elevation background site to provide insight into plume chemistry; and it makes use of photochemical modelling to establish baseline air quality conditions in the absence of any wildfire emissions to determine the smoke plume's contribution to degraded air quality and exceedances of regional air quality objectives and national standards of $O_3$ and $PM_{2.5}$.*"
>
> Removed text on ln15-18, pg 3
>
> Also noticed some redundancy in introduction text on ln 7-8, pg3 and it was revised to:
> "*Aerosol backscatter measurement and low depolarization volume ratios during the event showed the progressive entrainment of smoke into the LFV through July 10$^{th}$, 2012 which coincided with the high $PM_{2.5}$ observed by the region's fixed air quality monitoring network.* "

3.) Page 8, line 14, "… since the size distribution data from 10 nm to 1000 microns, measured by OPC,…" has limits incorrect for an OPC. Likely the 10 nm refers to an SMPS instrument, which was not working for some or most of the time under study…

> Author's response: ln 14, pg8 revised to:
>
> *"PM$_1$ mass is estimated from the ACSM, since the size distribution data indicate that most of the mass is below 0.7 μm; in Figure S3, an example is shown for July 9$^{th}$, 2012 when the SMPS became operational again."*

4.) page 11 line 19, "Fine organic aerosol mass (~88% based on the ACSM measurements)…." should include "mass fraction" in parentheses to be clear that 88% of the mass of the particles were organic.

> Author's response: ln 19, pg 11 revised to:
>
> *"Fine organic aerosol mass (based on the ACSM measurements) accounted for the majority (~88% by mass fraction) of the 1-hr and 24-hr PM$_{2.5}$ enhancements of 23 μg/m$^3$ and 10 μg/m$^3$, respectively."*